# Dissecting the phenotypic and functional heterogeneity of mouse inflammatory osteoclasts by the expression of *Cx3cr1*

**Maria-Bernadette Madel[1,2], Lidia Ibáñez[3], Thomas Ciucci[4], Julia Halper[1,2], Matthieu Rouleau[1,2], Antoine Boutin[1,2], Christophe Hue[5], Isabelle Duroux-Richard[6], Florence Apparailly[6], Henri-Jean Garchon[5,7], Abdelilah Wakkach[1,2], Claudine Blin-Wakkach[1,2]\***

[1]Laboratoire de PhysioMédecine Moléculaire, CNRS, Nice, France; [2]Université Côte d'Azur, Nice, France; [3]Department of Pharmacy, Cardenal Herrera-CEU University, Valencia, Spain; [4]Laboratory of Immune Cell Biology, Center for Cancer Research, National Cancer Institute, National Institutes of Health, Bethesda, United States; [5]Université Paris-Saclay, UVSQ, INSERM, Infection et inflammation, Montigny-Le-Bretonneux, France; [6]IRMB, Univ Montpellier, INSERM, CHU Montpellier, Montpellier, France; [7]Genetics division, Ambroise Paré Hospital, AP-HP, Boulogne-Billancourt, France

**\*For correspondence:**
claudine.BLIN@unice.fr

**Competing interests:** The authors declare that no competing interests exist.

**Abstract** Bone destruction relies on interactions between bone and immune cells. Bone-resorbing osteoclasts (OCLs) were recently identified as innate immune cells activating T cells toward tolerance or inflammation. Thus, pathological bone destruction not only relies on increased osteoclast differentiation, but also on the presence of inflammatory OCLs (i-OCLs), part of which express *Cx3cr1*. Here, we investigated the contribution of mouse Cx3cr1[+] and Cx3cr1[neg] i-OCLs to bone loss. We showed that Cx3cr1[+] and Cx3cr1[neg] i-OCLs differ considerably in transcriptional and functional aspects. Cx3cr1[neg] i-OCLs have a high ability to resorb bone and activate inflammatory CD4[+] T cells. Although Cx3cr1[+] i-OCLs are associated with inflammation, they resorb less and have in vitro an immune-suppressive effect on Cx3cr1[neg] i-OCLs, mediated by PD-L1. Our results provide new insights into i-OCL heterogeneity. They also reveal that different i-OCL subsets may interact to regulate inflammation. This contributes to a better understanding and prevention of inflammatory bone destruction.

## Introduction

Chronic inflammation and bone destruction are frequently associated due to complex interactions between activated immune cells and progenitors of bone-resorbing osteoclasts (OCLs). While they have long been attributed to the sole effect of immune cells on osteoclastogenesis, recent evidences demonstrated that these interactions are reciprocal (*Buchwald et al., 2012*; *Ibáñez et al., 2016*; *Kiesel et al., 2009*; *Li et al., 2010*). Indeed, OCLs are monocytic cells responding to immune signals and beside bone resorption, they present antigens and activate T cells (*Madel et al., 2019*). However, to date, the mechanisms that reciprocally link inflammation and bone destruction remain poorly understood.

OCLs are long-lived multinucleated cells arising from bone marrow (BM) progenitors (*Arai et al., 1999*; *Jacome-Galarza et al., 2013*; *Jacome-Galarza et al., 2019*). However, in inflammation, they also arise from inflammatory Ly6C[hi] monocytes (MNs) (*Ammari et al., 2018*; *Seeling et al., 2013*) and dendritic cells (DCs) including in vivo (*Wakkach et al., 2008*; *Ibáñez et al., 2016*;

*Mansour et al., 2012*; *Rivollier et al., 2004*; *Madel et al., 2019*). Furthermore, OCLs permanently fuse with MNs and undergo fission to maintain their nuclei number (*Jacome-Galarza et al., 2019*). We and others showed that depending on their origin and environment, OCLs induce different T cell responses. In steady state or when derived from BM MNs of healthy mice, OCLs activate regulatory CD4[+] and CD8[+] T cells (tolerogenic-OCLs/t OCLs) (*Ibáñez et al., 2016*; *Kiesel et al., 2009*), whereas when derived from DCs or during inflammation, they induce TNFα-producing CD4[+] T cells (inflammatory-OCLs/i-OCLs) (*Ibáñez et al., 2016*; *Madel et al., 2019*). Thus, according to their origin and environment, OCLs are heterogeneous and initiate different immune responses. However, the full contribution of i-OCLs to inflammatory bone loss remains to be elucidated.

Recently, we identified the fractalkine receptor Cx3cr1 as a marker identifying i-OCLs (*Ibáñez et al., 2016*). Cx3cr1 is expressed by OCL precursors and mediates their BM recruitment by Cx3cl1-producing endothelial cells and osteoblasts (*Han et al., 2014*; *Hasegawa et al., 2019*; *Koizumi et al., 2009*; *Matsuura et al., 2017*). During osteoclastogenesis, the level of expression of *Cx3cr1* decreases and is very low in t-OCLs (*Hoshino et al., 2013*; *Koizumi et al., 2009*). However, in inflammatory conditions, *Cx3cr1* remains expressed in about 25% of i-OCLs (*Ibáñez et al., 2016*). Nevertheless, the role of Cx3cr1 in i-OCLs as well as the function and contribution of Cx3cr1[neg] and Cx3cr1[+] i-OCLs in inflammation have not yet been explored.

We addressed these questions using *Cx3cr1*[GFP] mutant mice in which the *Gfp* gene is inserted in the second exon of the *Cx3cr1* gene (*Jung et al., 2000*). Consequently, while Cx3cr1 activity is normal in wild type (WT) *Cx3cr1*[GFP/+] mice, this protein is non-functional in *Cx3cr1*[GFP/GFP] mice and in both mice, GFP expression allows to identify Cx3cr1-expressing cells (*Jung et al., 2000*). We evaluated the consequences of *Cx3cr1* deficiency on inflammatory bone loss in vivo in ovariectomized (OVX) mice, a model where osteoclastogenesis is driven by TNF-α and RANK-L-producing CD4[+] T cells (*Cenci et al., 2000*; *Li et al., 2011*; *Weitzmann and Pacifici, 2006*). These conditions are similar to those priming Cx3cr1[+] OCLs previously described in inflammatory colitis (*Ibáñez et al., 2016*). We also compared Cx3cr1[+] and Cx3cr1[neg] i-OCLs as well as *Cx3cr1*-deficient i-OCLs by RNA-sequencing (RNA-seq) analysis and in vitro functional assays. We showed that Cx3cr1 deficiency does not change the bone resorption and T cell activation capacity of mature Cx3cr1[+] i-OCLs demonstrating that the Cx3cr1 protein per se does not play a major role in these functions. Furthermore, we showed that Cx3cr1 is a marker that identifies two distinct subsets of i-OCLs (Cx3cr1[neg] and Cx3cr1[+]) having different immune functions. Our findings unveil the heterogeneity of i-OCLs and their contribution to inflammation and bone loss. These new insights into osteoimmunology and OCL heterogeneity contribute to a better understanding of the bone microenvironment regulation and the underlying molecular processes during inflammatory bone destruction.

## Results

### Cx3cr1 deficiency protects against inflammatory bone loss in osteoporosis

We addressed in vivo the participation of Cx3cr1 in inflammatory osteoclastogenesis in ovariectomized (OVX) mice. We previously showed that Cx3cr1[+] i-OCLs increased in inflammatory colitis associated with bone destruction (*Ibáñez et al., 2016*) and we confirmed here that they also increased after ovariectomy in WT OVX mice (*Figure 1A*). To evaluate the implication of Cx3cr1 in OVX-induced bone loss, we used Cx3cr1-deficient *Cx3cr1*[GFP/GFP] and WT *Cx3cr1*[GFP/+] control mice. Micro-CT analysis revealed that after ovariectomy, Cx3cr1-deficient mice displayed moderate but significant higher BV/TV and trabecular number and lower trabecular separation than WT *Cx3cr1*[GFP/+] mice (*Figure 1B–C*). They also had less TRAcP[+] OCLs compared to OVX WT *Cx3cr1*[GFP/+] mice (*Figure 1D*). Thus, Cx3cr1 deficiency partially protects against bone destruction in OVX mice.

As Cx3cr1 is involved in the recruitment of OCL progenitors to the BM particularly under inflammatory conditions (*Charles et al., 2012*), we examined i-OCL progenitors in the BM of OVX Cx3cr1-deficient *Cx3cr1*[GFP/GFP] mice and WT *Cx3cr1*[GFP/+] mice. While both DCs and Ly6C[hi] MNs have been involved in the formation of OCLs in pathological conditions, only DCs were shown to give rise to Cx3cr1[+] i-OCLs (*Ibáñez et al., 2016*). Thus, we analyzed the osteoclastogenic capacity of blood Ly6C[hi] MNs sorted from *Cx3cr1*[GFP/+] mice. About 35% of the resulting OCLs expressed GFP

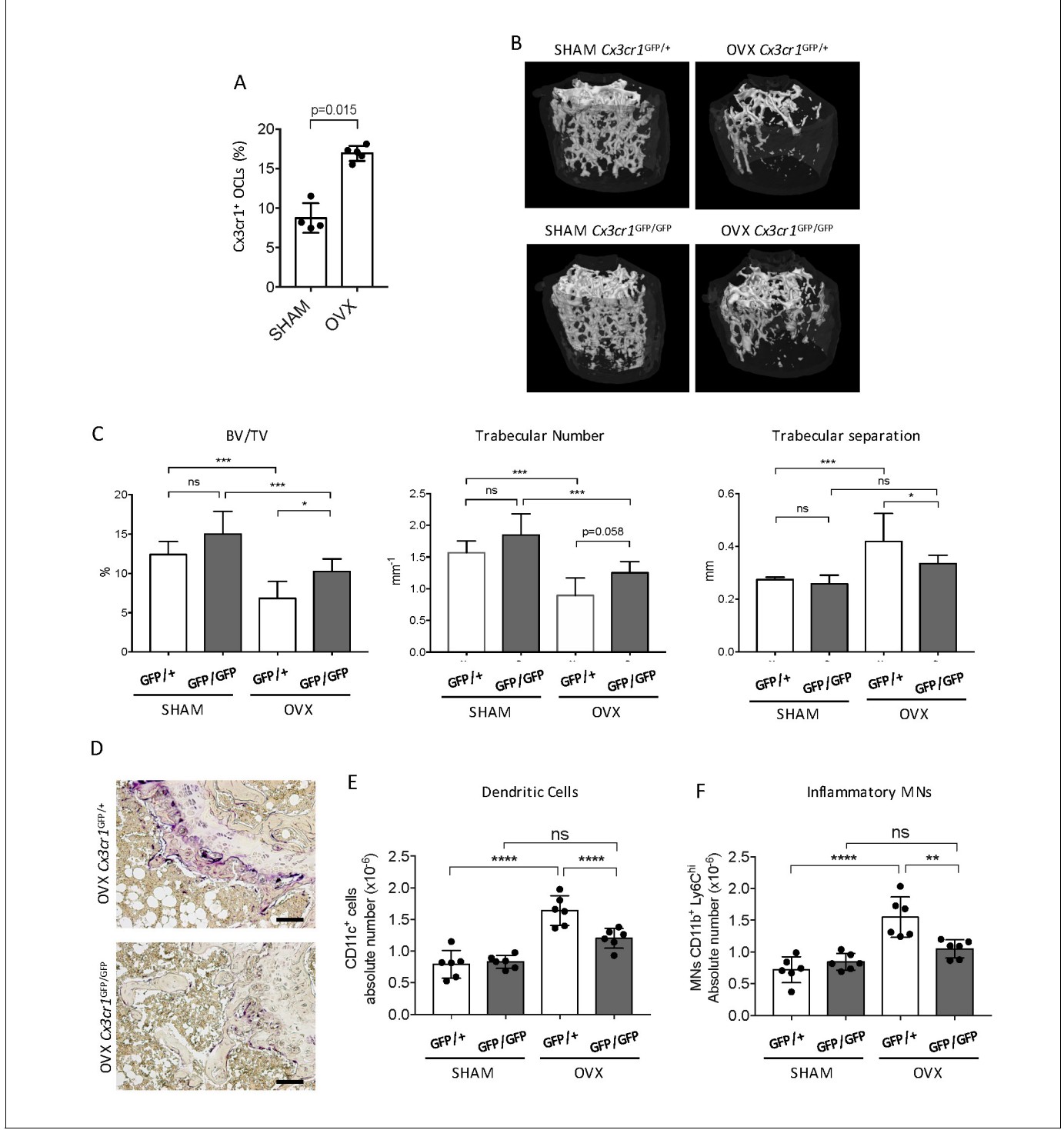

**Figure 1.** *Cx3cr1*-deficient mice display reduced bone loss after ovariectomy. (**A**) FACS analysis of Cx3cr1+ OCLs generated from BM cells from SHAM-operated and OVX WT mice. Histograms indicate the mean ± SD percentage of Cx3cr1+ i-OCLs (n = 4–5). (**B**) Representative images of femur μCT analysis from SHAM-operated and OVX WT *Cx3cr1*GFP/+ and Cx3cr1-deficient *Cx3cr1*GFP/GFP mice 6 weeks post-surgery. (**C**) Histograms show the mean ± SD percentage of bone volume fraction (BV/TV), trabecular number and trabecular separation (n = 8 mice per group). (**D**) Representative TRAcP staining on femora from OVX Cx3cr1GFP/+ and Cx3cr1GFP/GFP mice. Scale bar = 100 μm. (**E**) Ex vivo FACS analysis of CD11c+ DCs and (**F**) CD11b+Ly6Chi MNs present in the BM of SHAM and OVX mice. Cells were gated as in *Figure 1—figure supplement 2* and histograms are showing the absolute cell number of these populations in the bone marrow. **p<0.01; ***p<0.001; n.s., no significant difference.

The online version of this article includes the following figure supplement(s) for figure 1:

*Figure 1 continued on next page*

**Figure supplement 1.** Analysis of i-OCLs progenitors in OVX mice.
**Figure supplement 2.** Gating strategy for ex vivo FACS analysis.

indicating that Ly6C$^{hi}$ MNs are also progenitors of Cx3cr1$^+$ OCLs (*Figure 1—figure supplement 1A–B*). FACS analysis in OVX mice revealed that both DCs and Ly6C$^{hi}$ MNs increased in the BM of OVX *Cx3cr1*$^{GFP/+}$ but not *Cx3cr1*$^{GFP/GFP}$ mice (*Figure 1E–F*, *Figure 1—figure supplement 2*). These results suggest that the reduced bone loss observed in Cx3cr1-deficient *Cx3cr1*$^{GFP/GFP}$ mice is, at least partly, related to a decrease in BM progenitors of Cx3cr1$^+$ OCLs, namely DCs and Ly6C$^{hi}$ MNs.

## Cx3cr1 deficiency does not alter immune and bone resorption function of Cx3cr1$^+$ i-OCLs

These results suggest a central role of Cx3cr1 in i-OCL differentiation. To further investigate underlying mechanisms, we performed the first comparative RNA-seq approach on GFP$^+$ mature i-OCLs from WT *Cx3cr1*$^{GFP/+}$ and Cx3cr1-deficient *Cx3cr1*$^{GFP/GFP}$ mice (*Figure 2A*). OCLs represent a small population of cells firmly attached to the bone that cannot be isolated ex vivo in sufficient number for transcriptomic and functional assays (*Madel et al., 2018*). Thus, i-OCLs were differentiated from BM-derived CD11c$^+$ DCs of WT *Cx3cr1*$^{GFP/+}$ and Cx3cr1-deficient *Cx3cr1*$^{GFP/GFP}$ mice as described (*Ibáñez et al., 2016*). Mature Cx3cr1$^+$ i-OCLs were then sorted based on their GFP expression and multinucleation according to *Madel et al. (2018)* (*Figure 2—figure supplement 1*).

Except for their *Cx3cr1* expression, no differences were detected between GFP$^+$ i-OCLs from Cx3cr1-deficient *Cx3cr1*$^{GFP/+}$ and WT *Cx3cr1*$^{GFP/GFP}$ mice suggesting that Cx3cr1 deficiency did not change the identity of i-OCLs (*Figure 2B*, *Figure 2—figure supplement 2A–B*). Similarly, functional in vitro assays revealed that regardless if they derived from Cx3cr1-deficient *Cx3cr1*$^{GFP/+}$ or WT *Cx3cr1*$^{GFP/GFP}$ mice, GFP$^+$ i-OCLs had equivalent resorption activity, antigen-presentation capacity or expression of MHC-II and co-stimulatory molecules (*Figure 2C–E*). In addition, we performed a T cell activation assay using carboxyfluorescein succinimidyl ester (CFSE)-labelled CD4$^+$ T cells from OT-II mice bearing a T cell receptor (TCR) specific for the immunodominant ovalbumin (OVA) peptide (*Barnden et al., 1998*). No differences between GFP$^+$ i-OCLs from *Cx3cr1*$^{GFP/GFP}$ versus *Cx3cr1*$^{GFP/+}$ mice were found in their T cell activation and polarization capacity (*Figure 2F–G*). Thus, our results indicate that the function of the Cx3xr1 protein per se is not essential for the resorption and immune function of mature *Cx3cr1*-expressing i-OCLs.

## Cx3cr1$^+$ and Cx3cr1$^{neg}$ i-OCLs are distinct populations

Although Cx3cr1 is important for i-OCL progenitors, its deficiency does not affect bone resorption or T cell activation by mature Cx3cr1-expressing i-OCLs. However, being expressed in part of i-OCLs, Cx3cr1 remains a marker for the heterogeneity of these i-OCLs. We explored this heterogeneity and in particular the specific role of Cx3cr1-expressing and Cx3cr1$^{neg}$ i-OCL subsets, comparing them using an RNA-seq approach. Mature Cx3cr1$^+$ and Cx3cr1$^{neg}$ i-OCLs were generated in vitro from BM-derived CD11c$^+$ DCs of WT *Cx3cr1*$^{GFP/+}$ mice and sorted based on their GFP expression and multinucleation (*Figure 3A*, *Figure 2—figure supplement 1*; *Madel et al., 2018*). A total of 1771 genes were significantly differentially expressed (adj.pVal <0.05, Log$_2$FC $\geq$ 1) between the 2 populations. Principal Component Analysis (PCA) of all detectable genes revealed that Cx3cr1$^+$ (GFP$^+$) and Cx3cr1$^{neg}$ (GFP$^{neg}$) i-OCLs were clustered into 2 distinct populations (*Figure 3B*), which was reinforced by volcano plot and hierarchical clustering analysis of the top 107 significantly differently expressed genes (adj.pVal <0.05 and Log$_2$FC $\geq$ 1) (*Figure 3C–D*). In addition to *Cx3cr1*, genes interacting with its functional pathway were also differentially expressed (*Figure 3—figure supplement 1A*). In particular, *Nr1d1*, a negative regulator of *Cx3cr1* (*Song et al., 2018*) was reduced whereas *Tlr4* that mediates *Cx3cr1* upregulation by LPS (*Panek et al., 2015*) was increased in Cx3cr1$^+$ versus Cx3cr1$^{neg}$ i-OCLs (*Figure 3—figure supplement 1B–C*). These data suggest that *Cx3cr1* expression is tightly controlled during i-OCL differentiation. This was further confirmed by showing that Cx3cr1$^+$ OCLs could be generated from either Cx3cr1$^+$ or Cx3cr1$^{neg}$ BM-derived DCs (*Figure 3—figure supplement 1D*). Thus, the expression of *Cx3cr1* in i-OCLs is not only inherited from their progenitors but may be acquired through a dynamic up-regulation during the

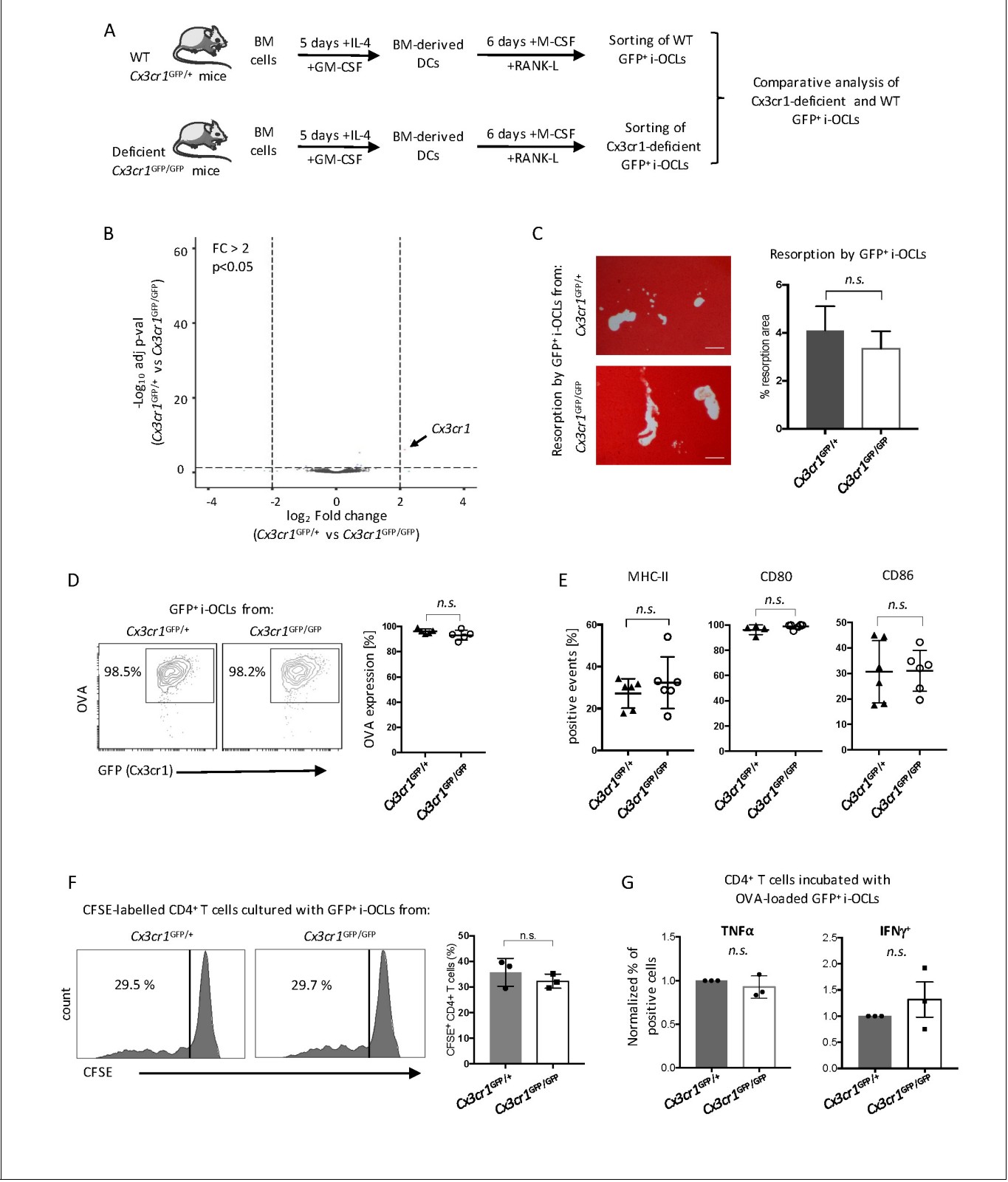

**Figure 2.** Cx3cr1 deficiency does not affect resorption and immune function in Cx3cr1+ i-OCLs. (**A**) Schematic representation of the differentiation of i-OCLs from BM-derived DCs of WT *Cx3cr1*GFP/+ and Cx3cr-deficient *Cx3cr1*GFP/GFP mice. (**B**) RNA-seq analysis on sorted GFP+ i-OCLs (representing Cx3cr1+ i-OCLs) differentiated from BM-derived DCs of WT *Cx3cr1*GFP/+ and Cx3cr-1deficient *Cx3cr1*GFP/GFP mice (gated as in *Figure 2—figure supplement 1*). Volcano-plot indicates –Log10 adjusted p value versus log2 fold-change comparing gene expression in i-OCL subsets. (**C**) Matrix

*Figure 2 continued on next page*

*Figure 2 continued*

dissolution activity of sorted GFP$^+$ i-OCLs from WT *Cx3cr1*$^{GFP/+}$ and Cx3cr1-deficient *Cx3cr1*$^{GFP/GFP}$ mice seeded at the same cell density on a calcified matrix was evidence by red alizarin staining of the mineralized matrix. Unstained areas correspond to the resorbed areas. Left panel: representative images of resorbed area. Scale bar = 100 μm. Right panel: quantification of resorbed areas presented as mean ± SD percentage of three independent biological replicates each in triplicates. (D) FACS analysis of fluorescent-OVA uptake among GFP$^+$ i-OCLs from *Cx3cr1*$^{GFP/+}$ and *Cx3cr1*$^{GFP/GFP}$ mice. Left panel: representative density plots and right panel: percentage of OVA$^+$ cells from five independent experiments. (E) FACS analysis of GFP$^+$ i-OCLs from *Cx3cr1*$^{GFP/+}$ and *Cx3cr1*$^{GFP/GFP}$ mice in at least four independent experiments. (F) T cell proliferation assay on CD4$^+$ T cells labelled with CFSE and cultured in the presence of OVA-challenged GFP$^+$ i-OCLs differentiated from *Cx3cr1*$^{GFP/+}$ and *Cx3cr1*$^{GFP/GFP}$ mice and analyzed by FACS after 4 days of coculture. (G) CD4$^+$ T cells activated by OVA-challenged GFP$^+$ i-OCLs from *Cx3cr1*$^{GFP/+}$ and *Cx3cr1*$^{GFP/GFP}$ mice were analyzed for their expression of TNFα and IFNγ by FACS after intracytoplasmic staining of these cytokines. n.s., no significant difference.

The online version of this article includes the following figure supplement(s) for figure 2:

**Figure supplement 1.** Gating strategy for mature Cx3cr1$^+$ and Cx3cr1$^{neg}$ i-OCLs.

**Figure supplement 2.** Comparative transcriptomic analysis of Cx3cr1$^+$ i-OCLs from *Cx3cr1*$^{GFP/GFP}$ and *Cx3cr1*$^{GFP/+}$ mice.

differentiation of a subset of i-OCLs having a specific function. In particular, genes and functional pathways differentially expressed between Cx3cr1$^+$ and Cx3cr1$^{neg}$ i-OCLs (*Figure 3D–E*) strongly suggested that the function of Cx3cr1$^+$ i-OCLs is related to the immune system. Therefore, we further characterized Cx3cr1$^+$ and Cx3cr1$^{neg}$ i-OCLs focusing on their bone resorption capacity and their immune potential.

Transcriptomic profiling revealed that Cx3cr1$^{neg}$ i-OCLs expressed higher levels of genes responsible for bone resorption (*Figure 4A*), which was confirmed by RT-qPCR (*Figure 4B*). We compared Cx3cr1$^{neg}$ and Cx3cr1$^+$ i-OCLs subsets to CD169$^+$CD68$^+$ macrophages that also express Cx3cr1 (*Figure 4—figure supplement 1A–B*). Our results revealed that both Cx3cr1$^{neg}$ and Cx3cr1$^+$ i-OCL subsets differed from BM-derived macrophages as they did not express CD68 and CD169 (*Figure 4—figure supplement 1C–D*). Moreover, the 2 i-OCLs subsets shared the main characteristics of OCLs, e.g. TRAcP expression and resorption activity (*Figure 4C–F*) while BM-macrophages did not express TRAcP (*Figure 4—figure supplement 1E*). However, when sorted mature Cx3cr1$^{neg}$ and Cx3cr1$^+$ i-OCLs were seeded at the same cell concentration, Cx3cr1$^{neg}$ i-OCLs showed a higher matrix dissolution activity compared to Cx3cr1$^+$ i-OCLs (*Figure 4E–F*). These results demonstrate that Cx3cr1$^{neg}$ and Cx3cr1$^+$ subsets correspond to 2 subsets of *bona fide* OCLs that differ in their resorption activity.

Besides these differences in bone resorption, RNA-seq data revealed significant differences in genes involved in antigen-processing and presentation (*Figure 5A*). Further in vitro analysis confirmed that whereas both i-OCL subsets engulfed OVA, this capacity was significantly decreased in Cx3cr1$^{neg}$ i-OCLs (*Figure 5B–C*). In line with the RNA-seq data, FACS analysis confirmed that MHC-II molecules were more expressed in Cx3cr1$^{neg}$ than Cx3cr1$^{pos}$ i-OCLs (*Figure 5D*), while no significant differences were observed in the expression of CD80 and CD86 (*Figure 5E–F*). Moreover, Cx3cr1$^{neg}$ i-OCLs showed higher capacity to induce T cell proliferation than Cx3cr1$^+$ i-OCLs (*Figure 5G*). Nevertheless, both i-OCL subsets were able to induce TNFα and IFNγ-producing CD4$^+$ T cells but not FoxP3$^+$CD4$^+$ regulatory T (Treg) cells (*Figure 5—figure supplement 1*), in accordance with the previously described immune properties of i-OCLs (*Ibáñez et al., 2016*).

## Cx3cr1$^+$ i-OCLs express immunosuppressive factors and control the immune function of Cx3cr1$^{neg}$ i-OCLs in vitro

RNA-seq data also revealed that both i-OCLs subsets expressed inflammation-associated genes in line with their capacity to induce TNFα and IFNγ-producing CD4$^+$ T cells (*Figure 6A*). However, Cx3cr1$^+$ i-OCLs had significant higher expression of immunosuppressive genes (*Figure 6A*) that could contribute to their lower inflammatory potential. Further RT-qPCR of the immunosuppressive genes *CD274* (PD-L1), *Lgals9* (Galectin-9) and *Tnfrsf14* (HEVM) confirmed these findings (*Figure 6B*). Interestingly, blocking the PD-L1/PD-1 axis with an anti-PD-1 antibody in co-cultures of Cx3cr1$^+$ i-OCLs and CD4$^+$ T cells increased the capacity of Cx3cr1$^+$ i-OCLs to induce T cell proliferation (*Figure 6C*). These findings reveal that PD-L1 plays a central role in the immunosuppressive potential of Cx3cr1$^+$ i-OCLs. They also suggest that Cx3cr1$^+$ i-OCLs could have an immunosuppressive effect on Cx3cr1$^{neg}$ i-OCLs. Therefore, we set up an immunosuppressive assay where OVA-loaded Cx3cr1$^{neg}$ i-OCLs were co-cultured with CFSE-labelled CD4$^+$ T cells from OT-II mice. Different ratios

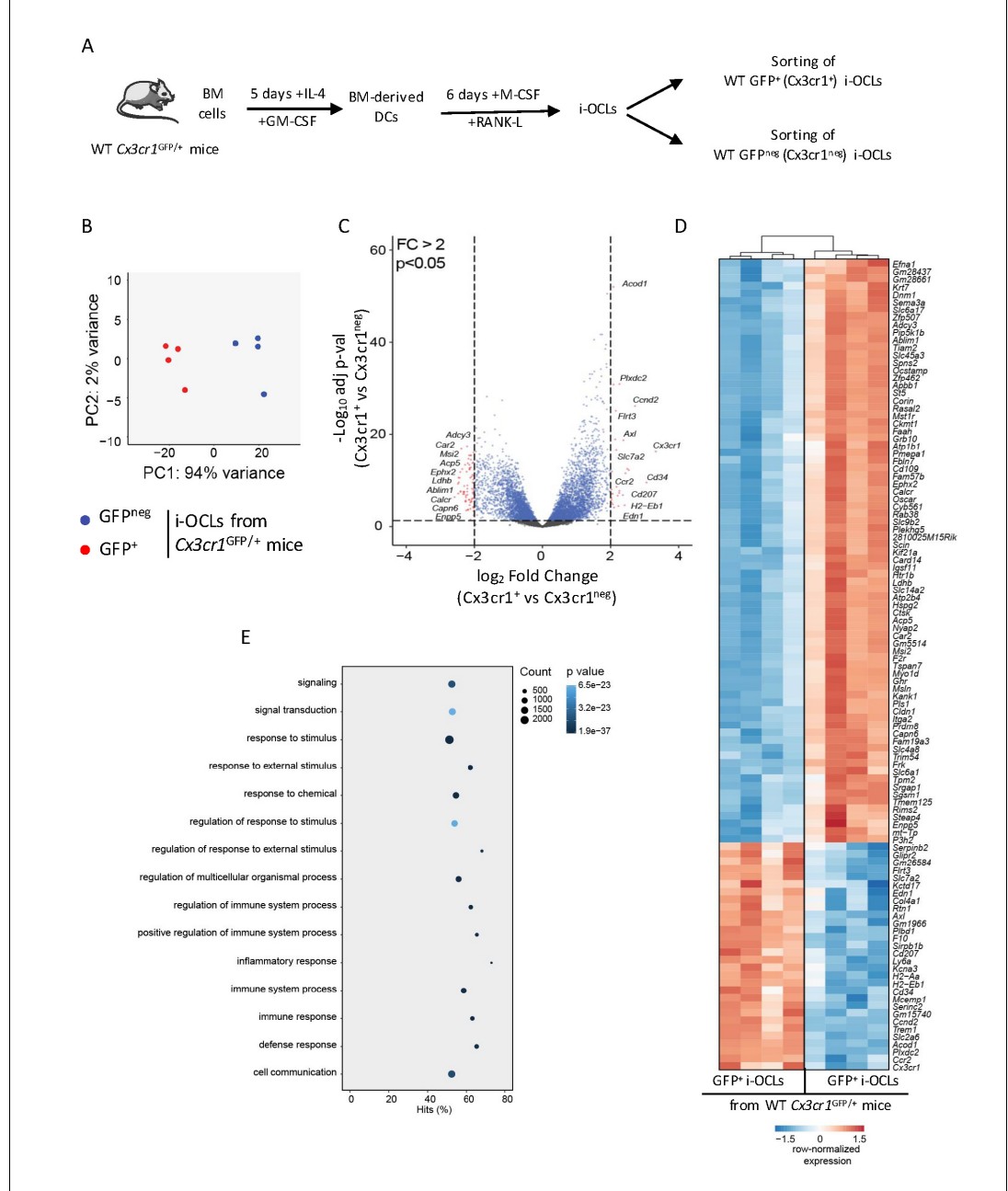

**Figure 3.** Transcriptomic profiling of Cx3cr1[+] and Cx3cr1[neg] i-OCLs reveals two distinct populations of i-OCLs. (**A**) Schematic representation of the differentiation of GFP[+] (Cx3cr1[+]) and GFP[neg] (Cx3cr1[neg]) i-OCLs from BM-derived DCs of WT *Cx3cr1*[GFP/+] mice and their sorting (gated as in *Figure 2—figure supplement 1*). (**B**) Principal component analysis of Cx3cr1[+] and Cx3cr1[neg] i-OCLs clusters samples in two groups (red: GFP[+] (Cx3cr1[+]) OCLs and blue GFP[neg] (Cx3cr1[neg]) OCLs, both from *Cx3cr1*[GFP/+] mice). Data shows the two first components on the top 500 most differentially expressed genes after batch correction. Each dot represents the expression profile of one sample. (**C**) Volcano-plot showing representative differentially expressed genes for Cx3cr1[+] and Cx3cr1[neg] i-OCLs. Cut-off values were defined by a fold change (FC) >2 and an adjusted p-value<0.05. (**D**) Heatmap visualization of the top 107 genes significantly differentially expressed between GFP[+] (Cx3cr1[+]) and GFP[neg] (Cx3cr1[neg]) i-OCLs from *Cx3cr1*[GFP/+] mice (adjusted p-value<0.05, FC ≥2). (**E**) Graphical representation of gene ontology analysis associated with differentially expressed genes between Cx3cr1[+] and Cx3cr1[neg] i-OCLs.

The online version of this article includes the following figure supplement(s) for figure 3:

**Figure supplement 1.** Cx3cr1[+] and Cx3cr1[neg] i-OCLs differ in their expression of Cx3cr1-interacting genes.

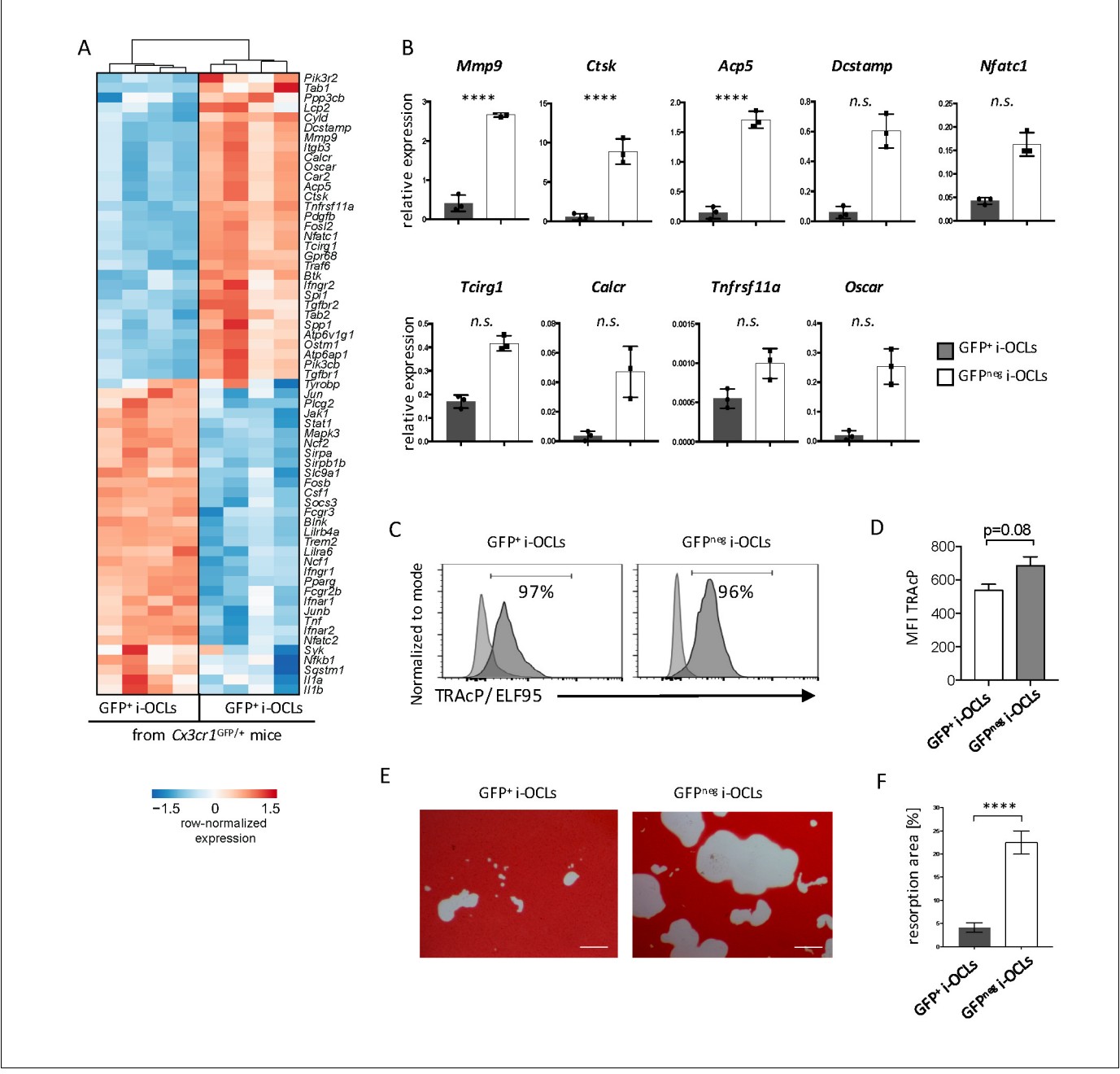

**Figure 4.** Cx3cr1neg and Cx3cr1+ i-OCLs differ in their resorbing activity. (**A**) Heatmap visualization of the z-scored expression for selected genes involved in bone resorption, osteoclast fusion and differentiation that are differentially expressed between GFP+ (Cx3cr1+) and GFPneg (Cx3cr1neg) i-OCLs both differentiated from WT *Cx3cr1*GFP/+ mice (adjusted p-value<0.05, FC ≥2). (**B**) RT-qPCR analysis on GFP+ (Cx3cr1+) and GFPneg (Cx3cr1neg) i-OCLs. Results are represented as the mean with 95% confidence interval of 3 independent biological replicates in triplicates. (**C**) Flow cytometry analysis of TRAcP expression using ELF95 substrate in the 2 i-OCLs subsets. Percentage of positive cells (dark curve) compared to the negative control (light curve) is indicated. (**D**) Quantification of the mean fluorescence intensity (MFI) of TRAcP in GFP+ (Cx3cr1+) and GFPneg (Cx3cr1neg) i-OCLs (n = 3). (**E**) Representative images of matrix dissolution activity of sorted GFP+ (Cx3cr1+) and GFPneg (Cx3cr1neg) i-OCLs seeded at the same cell density. Red alizarin staining of the mineralized matrix revealed the resorbed areas as unstained. Scale bar = 100 µm. (**F**) Quantification of resorbed areas from three independent experiments in triplicates are indicated as percentage. ****p<0.0001; n.s., no significant difference.

The online version of this article includes the following figure supplement(s) for figure 4:

**Figure supplement 1.** Cx3cr1+ i-OCLs differ from Cx3cr1+ BM-derived macrophages.

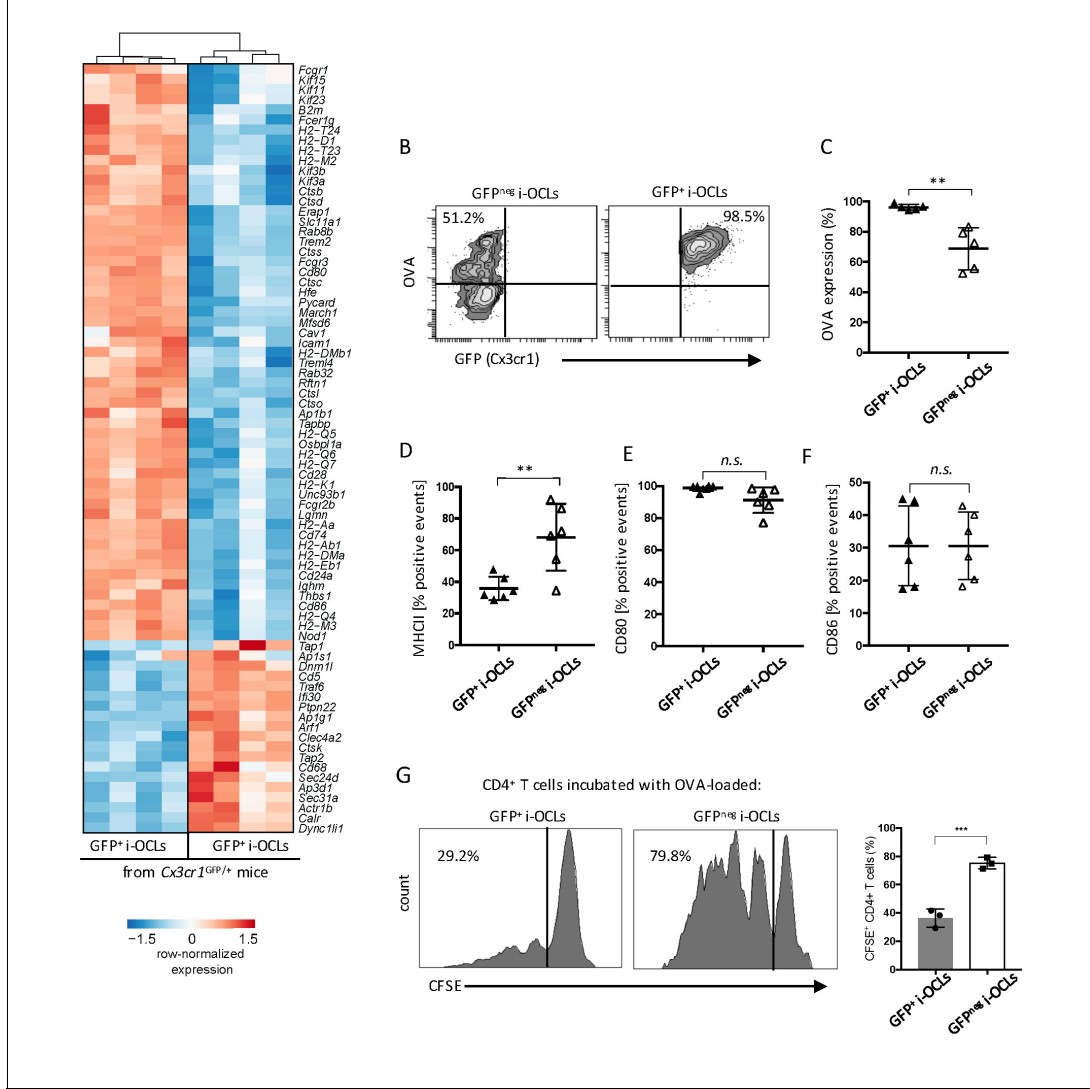

**Figure 5.** Cx3cr1[neg] and Cx3cr1[+] subsets differ in their antigen uptake and presentation. (**A**) Heatmap visualization of the z-scored expression for selected genes involved in antigen uptake, processing and presentation that are differentially expressed between GFP[+] (Cx3cr1[+]) and GFP[neg] (Cx3cr1[neg]) i-OCLs from WT *Cx3cr1*[GFP/+] mice (adjusted pVal <0.05, FC ≥2). (**B**) Representative FACS plots of OVA uptake in Cx3cr1[+] and Cx3cr1[neg] i-OCLs. (**C**) Quantification of FACS analysis for OVA uptake in GFP[+] (Cx3cr1[+]) and GFP[neg] (Cx3cr1[neg]) i-OCLs (n = 5). (**D–F**) FACS analysis of GFP[+] (Cx3cr1[+]) and GFP[neg] (Cx3cr1[neg]) i-OCLs for (**D**) MHC-II, (**E**) CD80 and (**F**) CD86 (n = 6). (**G**). Representative FACS histograms for T cell proliferation assay of CFSE-labelled CD4[+] T cells from OT-II mice cultured with OVA-loaded GFP[+] (Cx3cr1[+]) or GFP [neg] (Cx3cr1[neg]) i-OCLs after 5 days of coculture. **p<0.01; n.s., no significant difference.

The online version of this article includes the following figure supplement(s) for figure 5:

**Figure supplement 1.** CD4[+] T cell polarization analysis CD4[+] T cells from OT-II mice were cocultured with OVA-challenged GFP[+] (Cx3cr1[+]) and GFP[neg] (Cx3cr1[neg]) i-OCLs from WT *Cx3cr1*[GFP/+] mice, respectively, and analyzed by intracytoplasmic staining for their (**A**) TNFα, (**B**) IFNγ and (**C**) FoxP3 expression using flow cytometry. n.s., no significant difference.

of Cx3cr1[+] i-OCLs not challenged with OVA (unable to directly activate OT-II T cells) were added to the culture (*Figure 6D*). Cx3cr1[neg] i-OCLs alone (ratio 1:0) highly stimulated CD4[+] T cell proliferation, which was dramatically decreased by the addition of Cx3cr1[+] i-OCLs (ratio 1:2; *Figure 6E*). These data demonstrate that Cx3cr1[+] i-OCLs are able to control the inflammatory activity of Cx3cr1[neg] i-OCLs in vitro, suggesting the existence of a novel mechanism of interaction between OCLs in inflammatory conditions that remains to be addressed in vivo.

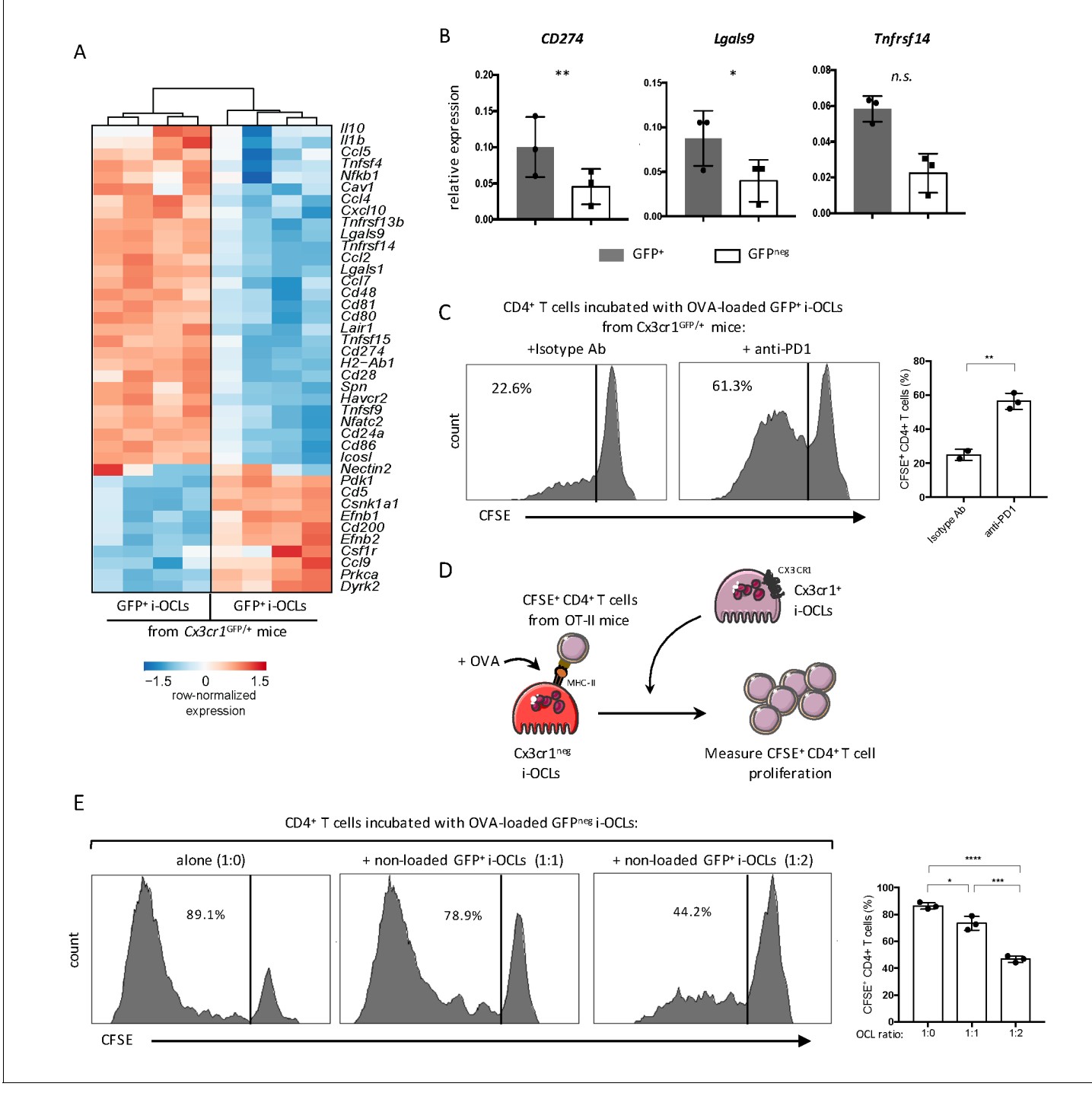

**Figure 6.** Cx3cr1[neg] and Cx3cr1[+] subsets differ in their T cell activation capacity. (**A**) Heatmap visualization of the z-scored expression for selected genes involved in T cell stimulation and inhibition that are differentially expressed between GFP[+] (Cx3cr1[+]) and GFP[neg] (Cx3cr1[neg]) i-OCLs from WT *Cx3cr1*[GFP/+] mice (adjusted p<0.05, FC ≥2). (**B**) RT-qPCR analysis of immunosuppressive molecules. Graphs show three independent experiments conducted in triplicates. (**C**) Representative FACS histograms and quantification of the proliferation of CSFE-labelled CD4[+] T cell from OT-II mice cocultured with OVA-loaded Cx3cr1[+] i-OCLs from WT *Cx3cr1*[GFP/+] mice and an isotype antibody (left panel) or an anti-PD-1 antibody (right panel). (**D**) Schematic representation of the experimental setup. Sorted GFP[neg] (Cx3cr1[neg]) i-OCLs from WT *Cx3cr1*[GFP/+] mice were loaded with OVA for 3 hr and incubated with CFSE[+] CD4[+] T cells in the presence of different rations of non-OVA loaded GFP[+](Cx3cr1[+]) i-OCLs from WT *Cx3cr1*[GFP/+] mice. (**E**) FACS analysis and quantification of CFSE[+] CD4[+] T cells of OT-II mice cocultured in the presence of different ratios (1:0; 1:1; 1:2) between OVA-loaded Cx3cr1[neg] i-OCLs and Cx3cr1[+] i-OCLs (non-loaded with OVA) for 5 days. *p<0.05; **p<0.01; ***p<0.001; ****p<0.0001; n.s., no significant difference.

## Discussion

This study provides further evidence of OCL heterogeneity. We previously demonstrated the existence of t-OCLs priming CD4[+] Treg cells and i-OCLs inducing TNFα[+]CD4[+] T cells that emerge under different conditions (*Ibáñez et al., 2016*). Here, we show that i-OCLs encompass two distinct populations of *bona fide* OCLs that can be distinguished by their *Cx3cr1* expression. Although these 2 i-OCL subsets prime TNFα-producing CD4[+] T cells and resorb, they display different capacities for bone resorption and immunosuppression. Moreover, they do not activate Treg cells, and thus, they differ considerably from t-OCLs (*Ibáñez et al., 2016*). We also show that Cx3cr1[+] i-OCLs control the inflammatory activity of Cx3cr1[neg] i-OCLs in vitro, suggesting that different OCL subsets could interact with each other to control their immune activity, an hypothesis that remains to be investigated in vivo (*Figure 7*).

Cx3cr1 is involved in the development and progression of chronic inflammatory diseases, such as colitis and arthritis (*Kotani et al., 2013*; *Rossini et al., 2014*; *Tarrant et al., 2012*) while Cx3cr1 deficiency reduces the severity of these diseases (*Niess and Adler, 2010*; *Rossini et al., 2014*; *Tarrant et al., 2012*). Our results extend the protective effect of Cx3cr1 deficiency to OVX-induced bone destruction. As in inflammatory bowel disease, this model is driven by TNFα and RANK-L-

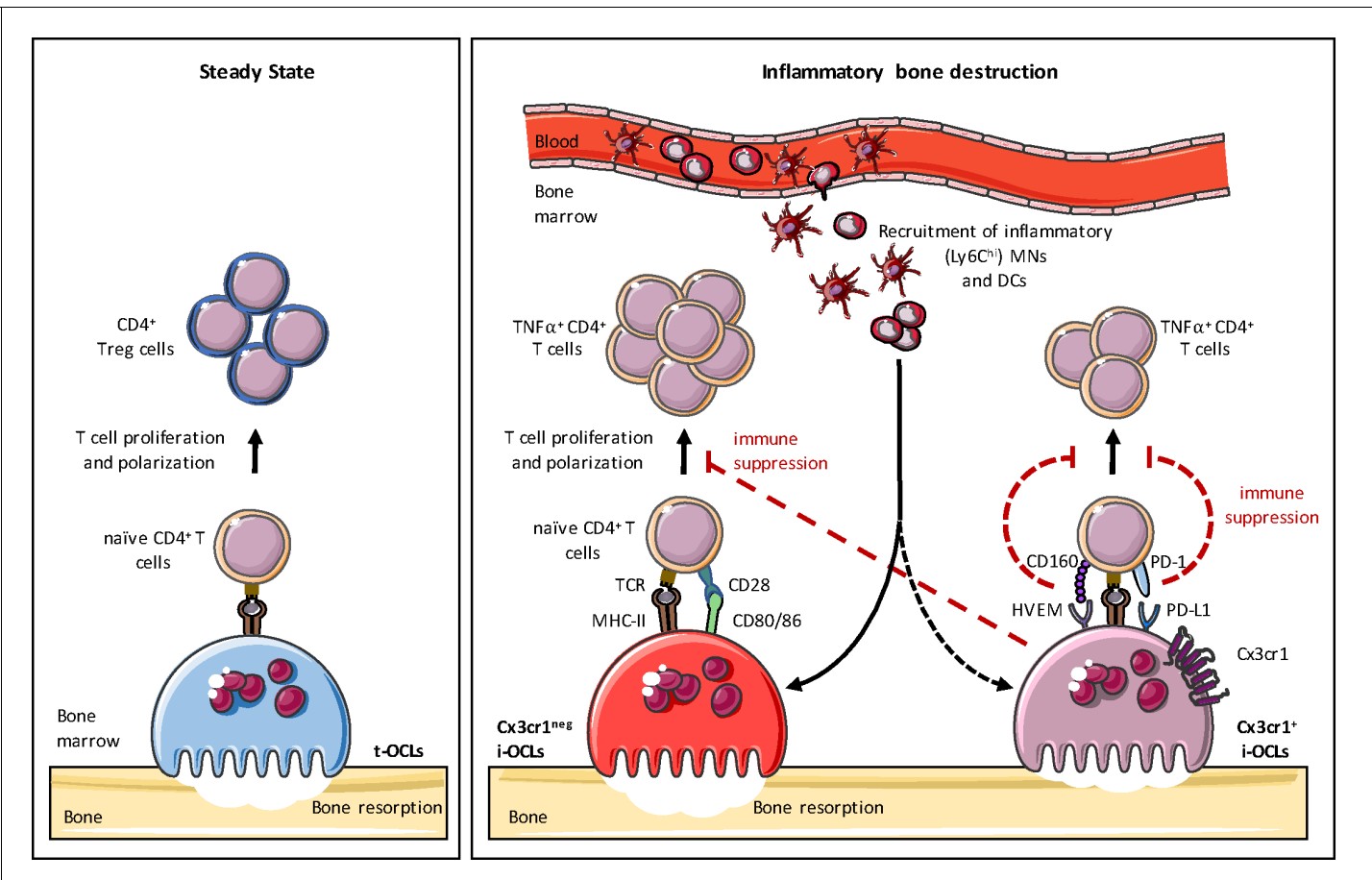

**Figure 7.** Heterogeneity of osteoclasts and underlying molecular mechanisms. In steady state, BM progenitors differentiate into tolerogenic OCLs (t-OCLs) that are able to present antigens and induce CD4[+] Treg cells. During inflammation, inflammatory MNs and DCs are recruited to the BM and differentiate into i-OCLs. Approx. 25% of these i-OCLs can be characterized by their expression of Cx3cr1 while the majority of i-OCLs does not express this marker. Cx3cr1[neg] i-OCLs show significantly higher bone resorption activity in vitro compared to Cx3cr1[+] i-OCLs. Both i-OCL subsets act as antigen-presenting cells but differ in their T cell activation capacity that is higher for the Cx3cr1[neg] i-OCL subset. Both i-OCL subsets are able to induce TNFα-producing CD4[+] T cells, however the Cx3cr1[+] OCL subset express high levels of co-inhibitory molecules such as PD-L1 that reduce their capacity to activate T cells. Moreover, Cx3cr1[+] i-OCLs have an immune suppressive effect on Cx3cr1 [neg] i-OCLs by reducing their T cell proliferation capacity in vitro.

producing CD4$^+$ T cells leading to the development of Cx3cr1$^+$ OCLs (*Cenci et al., 2000*; *Ibáñez et al., 2016*; *Weitzmann and Pacifici, 2006*). As a chemokine receptor, Cx3cr1 is required for the migration of monocytic cells and was suggested to participate in the maintenance of OCL progenitors particularly in inflammatory conditions (*Charles et al., 2012*; *Han et al., 2014*; *Hasegawa et al., 2019*). Its ligand (Fractalkine/Cx3cl1) increases in OVX mice and osteoporotic patients and correlates with disease severity (*Chen et al., 2016*; *Wang et al., 2017*). Fractalkine-producing mesenchymal cells attract OCL precursors and support their differentiation (*Goto et al., 2016*; *Matsuura et al., 2017*). In agreement with these data, we demonstrate that the reduced bone loss in OVX Cx3cr1-deficient mice is related to a decrease in BM i-OCL precursors (DCs and Ly6C$^{hi}$ MNs) and a reduced osteoclastogenesis. Although Cx3cr1-deficient mice were reported to have slightly reduced bone density (*Hoshino et al., 2013*), we only observed moderate but significant differences in OVX mice, highlighting the importance of Cx3cr1 in pathological bone destruction. Compared to previous reports (*Hoshino et al., 2013*), we gained advantage from working on purified mature i-OCLs from Cx3cr1-deficient mice and we did not observe any significant variations when comparing Cx3cr1$^+$ i-OCLs from Cx3cr1-deficient *Cx3cr1*$^{GFP/GFP}$ and WT *Cx3cr1*$^{GFP/+}$ mice. Overall, our results demonstrate that the Cx3cr1/Cx3cl1 axis is essential for the increased BM recruitment and differentiation of i-OCL progenitors observed in OVX mice but that the Cx3cr1 protein *per se* is not involved in the resorption and immune function of fully mature i-OCLs.

This raises the question of the role of Cx3cr1 in a subset of mature i-OCLs. As OCLs permanently fuse with new monocytic cells (*Jacome-Galarza et al., 2019*), Cx3cr1 could be considered as a marker for some of these progenitor cells that remains expressed in the resulting OCLs. However, Cx3cr1$^+$ OCLs are induced by inflammatory cytokines such as IL-17 and TNF$\alpha$ and arise in conditions associated with inflammatory bone destruction (*Ibáñez et al., 2016*; *Madel et al., 2019*). Furthermore, Cx3cr1$^+$ and Cx3cr1$^{neg}$ i-OCLs significantly differ in the expression of genes involved in Cx3cr1-interacting pathways. For instance, Cx3cr1$^+$ OCLs show lower expression of *Nr1d1*, a Cx3cr1 inhibitor (*Song et al., 2018*) and higher levels of *Tlr4* that mediates the positive effects of LPS on Cx3cr1 (*Panek et al., 2015*). Furthermore, we observed that Cx3cr1$^+$ OCLs arise from both Cx3cr1$^+$ and Cx3cr1$^{neg}$ progenitors (*Figure 3—figure supplement 1D*), indicating that *Cx3cr1* expression by some mature OCLs is inducible and not solely inherited from their progenitors.

In line with our observations in OCLs, Cx3cr1 is also a marker for MN and DC diversity (*Ginhoux et al., 2009*; *Varol et al., 2009*; *Yona et al., 2013*). Depending on their environment, BM progenitors can differentiate into immunogenic Cx3cr1$^{low/neg}$MHC-II$^{hi}$ DCs or Cx3cr1$^+$MHC-II$^{low}$ macrophages with high antigen-uptake and a PD-L1-associated immunosuppressive function (*Zhang et al., 2012*). In the gut mucosa, Cx3cr1$^{hi}$ macrophages efficiently take up antigens but induce lower T cell proliferation compared to CD103$^+$Cx3cr1$^{neg}$ DCs (*Schulz et al., 2009*). The balance between immunosuppressive Cx3cr1$^{hi}$ and inflammatory Cx3cr1$^{low}$ phagocytes in the gut is essential in controlling inflammatory responses in colitis (*Regoli et al., 2017*; *Yin et al., 2012*). Accordingly, Cx3cr1$^+$ and Cx3cr1$^{neg}$ i-OCLs exert different immune functions. Cx3cr1$^+$ cells were more efficient in antigen uptake but less potent in T cell activation than Cx3cr1$^{neg}$ i-OCLs. In addition, as Cx3cr1$^{hi}$ macrophages, Cx3cr1$^+$ i-OCLs decrease T cell proliferation, notably via immunosuppressive molecules such as PD-L1, Galectin-9 and HVEM. These factors are major immune checkpoints involved in autoimmune disorders and tumor-induced immune suppression associated with T cell dysfunction, exhaustion, and low activation (*Bjordahl et al., 2013*; *Cai and Freeman, 2009*; *Francisco et al., 2009*; *Tai et al., 2018*). Thus, their high expression in Cx3cr1$^+$ i-OCLs further strengthens the hypothesis that Cx3cr1$^+$ i-OCLs act as immunosuppressive cells emerging upon inflammatory signals to regulate inflammation, which was confirmed in vitro by a functional immunosuppressive assay (*Figure 6*).

The contribution of OCLs to an immunosuppressive BM microenvironment has been recently explored in the context of multiple myeloma (MM). Bone destruction in MM is supported by malignant plasma cells that favor OCL differentiation, including from DCs, through high RANK-L, IL-17 and TNF$\alpha$ production (*Mansour et al., 2017*; *Noonan et al., 2010*; *Tucci et al., 2011*). In these conditions, OCLs suppress T cell activation via PD-L1, Galectin-9 and HVEM (*An et al., 2016*; *Mansour et al., 2017*; *Tai et al., 2018*). Although OCL heterogeneity was not investigated in MM, osteoclastogenic conditions are similar to those supporting the differentiation of Cx3cr1$^+$ i-OCLs (*Ibáñez et al., 2016*). Thus, our results provide new insights into the immunosuppressive property of

OCLs and suggest that Cx3cr1[+] i-OCLs are likely to be involved in pathological conditions such as MM.

Although Cx3cr1[+] and Cx3cr1[neg] i-OCLs both induce TNFα-producing CD4[+] T cells and are resorbing cells, we showed that mature Cx3cr1[neg] i-OCLs are more efficient to induce CD4[+] T cell proliferation and have a higher resorption capacity in vitro than Cx3cr1[+] i-OCLs. This suggests that Cx3cr1[neg] i-OCLs play a major role in inflammatory bone destruction such as in osteoporosis and chronic inflammatory diseases. Interestingly, when comparing Cx3cr1[+] i-OCLs from Cx3cr1-deficient *Cx3cr1*[GFP/GFP] and WT *Cx3cr1*[GFP/+] mice no differences were observed. Thus, the Cx3cr1 protein as such is not essential for the immunomodulatory and bone resorbing function of Cx3cr1[+] i-OCLs. However, it remains a membrane marker allowing to distinguish between 2 i-OCLs subsets (namely Cx3cr1[+] and Cx3cr1[neg] i-OCLs) exerting different immunomodulatory roles. Apart from the lack of Cx3cr1 expression and their immunogenic effect, there is currently no specific marker for the identification of Cx3cr1[neg] i-OCLs. Thus, an in-depth characterization of these cells is necessary to fully address their specific role. Interestingly, Cx3cr1[neg] i-OCLs induce high proportions of TNFα and IFNγ-producing CD4[+] T cells in vitro. These cytokines are potent inducers of Cx3cr1 and PD-L1 (*Bjordahl et al., 2013*; *McGrath and Najafian, 2012*; *Tsukamoto et al., 2019*) suggesting that TNFα-producing T cells induced by Cx3cr1[neg] i-OCLs may participate in the regulation of Cx3cr1[+] i-OCLs. This observation supports the existence of a regulatory loop between pro-inflammatory Cx3cr1[neg] i-OCLs and immunosuppressive Cx3cr1[+] i-OCLs that negatively regulate Th1 cell activation to control inflammation and that remains to be confirmed in vivo. Therefore, our findings suggest that the balance between the 2 i-OCL subsets would be an important aspect of the immune regulation during inflammatory bone loss.

In summary, we describe 2 OCL subsets that arise during inflammation and that are characterized by their *Cx3cr1* expression (*Figure 7*). While Cx3cr1[neg] i-OCLs mediate inflammatory bone destruction, Cx3cr1[+] i-OCLs are more prone to immunosuppression. This might contribute to the regulation of inflammatory processes but also to maintain an immunosuppressive BM microenvironment. Therefore, our in vitro data suggest the existence of a negative feedback mechanism by a gatekeeper subset of *Cx3cr1*-expressing i-OCLs during inflammation. This strongly emphasizes the heterogeneity of i-OCLs and provides new insights in their inflammatory function. Our study uncovered a hitherto neglected OCL diversity that contributes to a deeper understanding of the balance between inflammation and immunosuppression in the BM. Interestingly, OCL-related reduction of T cell activation can be restored using anti-PD-1 antibodies. Thus Cx3cr1[+] i-OCLs might represent a novel promising target to overcome the OCL-mediated immunosuppression and restore an anti-tumorigenic BM microenvironment. On the other hand, our results suggest that Cx3cr1[neg] i-OCLs are promising targets against inflammatory bone destruction. A deeper comprehension of OCL diversity is therefore indispensable to identify more specific markers and to evaluate the therapeutic interest of targeting specific OCL subsets.

# Materials and methods

## Key resources table

| Reagent type (species) or resource | Designation | Source or reference | Identifiers | Additional information |
|---|---|---|---|---|
| Strain, strain background (*Mus musculus*) | Cx3cr1[GFP/+] | *Jung et al., 2000* | | CDTA, CNRS, Orléans, France |
| Sequenced-based reagent | *Calcr* | | PCR primers | CTTCCATGCTGATCTTCTGG and CAGATCTCCATTGGGCACAA |
| Sequenced-based reagent | *Acp5* | | PCR primers | TGCCTACCTGTGTGGACATGA and CACATAGCCCACACCGTTCTC |
| Sequenced-based reagent | *Mmp9* | | PCR primers | TGAGTCCGGCAGACAATCCT and CGCCCTGGATCTCAGCAATA |

*Continued on next page*

*Continued*

| Reagent type (species) or resource | Designation | Source or reference | Identifiers | Additional information |
|---|---|---|---|---|
| Sequenced-based reagent | *Ctsk* | | PCR primers | CAGCAGAGGTGTGTACTATG and GCGTTGTTCTTATTCCGAGC |
| Sequenced-based reagent | *Atp6V0a3/Tcirg1* | | PCR primers | CGCTGCGAGGAACTGGAG and AGCGTCAGACCTGCCCG |
| Sequenced-based reagent | *Tnfrsf11a* | | PCR primers | CTTGGACACCTGGAATGAAGAAG and AGGGCCTTGCCTGCATC |
| Sequenced-based reagent | *Nfatc1* | | PCR primers | TGAGGCTGGTCTTCCGAGTT and CGCTGGGAACACTCGATAGG |
| Sequenced-based reagent | *Dcstamp* | | PCR primers | GGGCACCAGTATTTTCCTGA and CAGAACGGCCAGAAGAATGA |
| Sequenced-based reagent | *Oscar* | | PCR primers | GTAACGGATCAGCTCCCCAG and CGCGGTACAGTGCAAAACTC |
| Sequenced-based reagent | *CD274* | | PCR primers | CAAGCGAATCACGCTGAAAG and GGGTTGGTGGTCACTGTTTGT |
| Sequenced-based reagent | *Lgals9* | | PCR primers | TCAAGGTGATGGTGAACAAGAAA and GATGGTGTCCACGAGGTGGTA |
| Sequenced-based reagent | *Lightr/Tnfrsf14* | | PCR primers | TGTCCATTCTTTTGCCACTTG and CCTGATGGTGTTCTCCTGTTGTT |
| Sequenced-based reagent | *36B4* | | PCR primers | TCCAGGCTTTGGGCATCA and CTTTATCAGCTGCACATCACTCAGA |
| Antibody | Monoclonal anti-mouse CD80 | BD Biosciences | clone 16-10A1 | APC-conjugated (1:100) |
| Antibody | Monoclonal anti-mouse CD86 | BD Biosciences | clone GL1 | PE-conjugated (1:100) |
| Antibody | Monoclonal anti-mouse I-A[b] | BD Biosciences | clone AF6-120.1 | Biotin-conjugated (1:200) |
| Antibody | Monoclonal anti-mouse/human CD11b | BD Biosciences | clone M1/70 | PE-conjugated (1:500) |
| Antibody | Monoclonal anti-mouse CD4 | BD Biosciences | clone RM4-5 | PECy7-conjugated (1:1000) |
| Antibody | Monoclonal anti-mouse CD68 | ThermoFisher Scientific | clone FA-11 | PECy7-conjugated (1:200) |
| Antibody | Monoclonal anti-mouse CD11c | ThermoFisher Scientific | clone N418 | PECy7-conjugated (1:200) |
| Antibody | Monoclonal anti-mouse Ly6C | ThermoFisher Scientific | clone HK1.4 | PerCP-conjugated (1:1000) |
| Antibody | Monoclonal anti-mouse CD169 | ThermoFisher Scientific | clone SER-4 | APC-conjugated (1:100) |

*Continued on next page*

*Continued*

| Reagent type (species) or resource | Designation | Source or reference | Identifiers | Additional information |
|---|---|---|---|---|
| Antibody | Monoclonal anti-mouse IFNγ | ThermoFisher Scientific | clone XMG1.2 | APC-conjugated (1:400) |
| Antibody | Monoclonal anti-mouse TNFα | ThermoFisher Scientific | clone MP6-XT22 | BV421-conjugated (1:400) |
| Chemical compound, drug | streptavidin | ThermoFisher Scientific | | PercP-conjugated (1:200) |
| Chemical compound, drug | ovalbumin | ThermoFisher Scientific | | Alexa Fluor 647-conjugated |
| Commercial assay or kit | Acid Phosphatase Leucocyte (TRAcP) kit | Sigma-Aldrich | 387A | |
| Commercial assay or kit | ELF 97 Endogenous Phosphatase Detection Kit | Molecular probes | E-6601 | |

## Mice and ovariectomy-induced osteoporosis

OT-II mice were purchased from Charles River Laboratory and C57BL/6 *Cx3cr1*GFP/+ mice were kindly provided by F. Laurent (INRA, Nouzilly, France). Animals were maintained under a 12 hr light/12 hr dark cycle with free access to water and standard mouse diet. For ovariectomy, female mice were randomly divided into 2 groups, which were then sham-operated (SHAM) or ovariectomized (OVX). Ovariectomy and sham surgery were performed on 6 weeks old *Cx3cr1*GFP/+ and *Cx3cr1*GFP/GFP female mice after anesthesia with isoflurane. The animals were intensively monitored for 72 hr after the procedure. Animals were weighed at the beginning of the study and twice a week thereafter. Six weeks after ovariectomy, mice were sacrificed. Uteri were weighed to control the quality of ovariectomy (not shown). Bones were used either for bone marrow cell isolation and further flow cytometry analysis or for subsequent µCT and histological analysis. Approval for animal experiments was obtained from the Institutional Ethics Committee on Laboratory Animals (CIEPAL-Azur, Nice Sophia-Antipolis, France) and experiments were conducted in compliance with ethical regulations for animal testing and research.

## Bone analyses

Femora were fixed in 4% paraformaldehyde (PFA) overnight. Subsequently, microcomputed tomography (µCT) was performed using the Skyscan 1176 µCT system (Bruker µCT, Belgium) at the preclinical platform ECELLFRANCE (IRMB, Montpellier, France). Scans were performed using isotropic voxels size of 18 µm, voltage of 50 kV, current of 500 mA, 0.5 mm aluminum filter, 180 degrees with a 0.7-degree rotation step and 210 ms exposure time. Data 3D reconstructions were generated for visual representation using NRecon software (Bruker µCT, Belgium).

For histological analysis, fixed femora were decalcified in 10% EDTA for 10 days, embedded in paraffin and tartrate-resistant acid phosphatase (TRAcP) staining was performed on 7 µm sections following manufacturer's recommendations (Sigma).

## Primary osteoclast culture

As OCLs cannot be isolated directly ex vivo in sufficient number for subsequent functional analysis, inflammatory OCLs (i-OCLs) were differentiated in vitro from BM-derived CD11c+ DCs or from blood CD11b+ Ly6Chi MNs of 6 week old WT *Cx3cr1*GFP/+ and deficient *Cx3cr1*GFP/GFP mice as previously described (*Ibáñez et al., 2016*). Briefly, bone marrow was flushed out of the long bones and after red blood cell lysis, BM cells were cultured in 24-well plates at $0.5 \times 10^6$ cells/well in RPMI medium (ThermoFisher Scientific) supplemented with 5% serum (Hyclone, GE Healthcare), 1% penicillin-streptomycin, 50 µM β-mercaptoethanol (both from ThermoFisher Scientific), 10 ng/ml IL-4 and 10 ng/ml

GM-CSF (both from PeproTech). CD11c$^+$ DCs were sorted using anti-CD11c antibody (1:200; HL3; BD Biosciences) and anti-biotin microbeads (Miltenyi Biotec). Alternatively, when indicated, i-OCLs were obtained from blood SSC$^{lo}$ CD11b$^+$ Ly6C$^{hi}$ MNs of *Cx3cr1*$^{GFP/+}$ OVX mice sorted on a FACS-Aria II (BD Biosciences) using anti-CD11b and Ly6C antibodies. For i-OCLs culture, $2 \times 10^4$ CD11c$^+$ DCs or CD11b$^+$ Ly6C$^{hi}$ MNs were seeded/well on 24-well plates in MEM-alpha (ThermoFisher Scientific) supplemented with 5% serum (Hyclone, GE Healthcare), 1% penicillin-streptomycin, 50 μM β-mercaptoethanol, 25 ng/ml M-CSF and 30 ng/ml RANK-L (both R and D). Cells were cultured at 37°C and 5% $CO_2$. Medium was changed every 3–4 days. Fully differentiated i-OCLs were stained with TRAcP (Sigma-Aldrich) according to manufacturer's instructions or collected from the culture plates using Accutase (Sigma-Aldrich) and used for subsequent FACS analysis or FACS sorting based on their nuclei number using H33342 (Sigma-Aldrich) as described previously (*Madel et al., 2018*). When indicated, macrophages were differentiated in vitro from total BM cells ($0.5 \times 10^6$ BM cells/ well in 24-well plates) in MEM-alpha supplemented with 5% serum (Hyclone, GE Healthcare), 1% penicillin-streptomycin, 50 μM β-mercaptoethanol and 25 ng/ml M-CSF for a total of 6 days.

## FACS and cell sorting on osteoclasts

Mature in vitro differentiated i-OCLs were analyzed and sorted based on their multinucleation and GFP expression as described previously (*Madel et al., 2018*) using flow cytometry analysis. Briefly, cells were incubated with 5 μg/ml H33342 (Sigma-Aldrich) in PBS 1X supplemented with 1% FBS and 2 mM EDTA (PSE) for 30 min at 37°C. For cell sorting, cells were filtered through a 100 μm nylon mesh and sorted on a FACS Aria IIu (BD Bioscience) using a 100 μm nozzle at a flow rate of 2000 events/s. Sorted cells were collected in FBS. Hoechst-stained i-OCLs from *Cx3cr1*$^{GFP}$ mutant mice were labeled with antibodies specific for murine anti-CD80, CD86 and IAb (MHC-II, biotinylated) antibodies with a PercP-conjugated streptavidin in PSE for 15 min on ice. For antigen-uptake, mature OCLs were collected and incubated in the presence of 25 μg/ml ovalbumin (ThermoFisher Scientific) for 2.5 hr at 37°C before adding 5 μg/ml H33342 staining for another 30 min and subsequent FACS acquisition. Gating strategies for Cx3cr1$^+$ and Cx3cr1$^{neg}$ i-OCLs are shown in *Figure 2—figure supplement 1*. For FACS analysis of TRAcP expression, OCLs were stained using the ELF97 substrate according to the manufacturer's protocol (endogenous phosphatase detection kit; Molecular Probes).

For ex vivo FACS analysis of OCL progenitors, after red blood cell lysis (Sigma-Aldrich) BM cells were labelled with anti-CD11c, CD11b and Ly6C antibodies. For FACS analysis of macrophages, BM-derived macrophages were labeled with anti-CD68 and CD169 antibodies.

For intracellular cytokine analysis, CD4$^+$ T cells were stimulated with PMA, ionomycin and Brefeldin A, labelled with an anti-CD4 antibody and fixed with 4% formaldehyde overnight as described (*Ciucci et al., 2015*). Cells were stained with anti-IFNγ and TNFα antibodies in Saponin 1X. Data were acquired using a FACS Canto-II (BD Biosciences).

## Resorption assay

Mature in vitro differentiated i-OCLs were sorted based on their multinucleation and GFP (Cx3cr1) expression as described above (*Figure 2—figure supplement 1*; *Madel et al., 2018*). A total of $10^3$ Cx3cr1$^+$ or Cx3cr1$^{neg}$ i-OCLs was seeded/well on a 96-well osteoassay plate (Corning) in MEM-alpha containing 10% FBS (Hyclone, GE Healthcare) and 30 ng/ml RANK-L (R and D). After 48 hr, matrix dissolution activity was evaluated by removing the cells using distilled sterile water and staining with 2% Alizarin Red (Sigma-Aldrich) for 1 min. Imaging was performed using an Axio Observer D1 microscope (Zeiss) and images were taken using AxioVision Rel. 4.8 software (Zeiss). Resorbed areas were quantified using Fiji/ImageJ software (*Schindelin et al., 2012*).

## T cell proliferation and immune suppression assay

Mature in vitro differentiated Cx3cr1$^+$ and Cx3cr1$^{neg}$ i-OCLs were sorted based on their multinucleation and GFP expression as mentioned above (*Figure 2—figure supplement 1*) and seeded on 96-well plates with $10^3$ cells/well in the presence of 600 nM ovalbumin (OVA) peptide (ThermoFisher Scientific). Spleens of 8 weeks old OT-II mice were used to isolate CD4$^+$ T cells that express transgenic OVA-specific αβ-TCRs by using the Dynabeads Untouched Mouse CD4 Cell Isolation Kit (ThermoFisher Scientific) according to manufacturer's specifications. To analyze T cell proliferation, CD4$^+$

T cells were labelled with 250 μM CFSE for 10 min at 37°C. OVA-loaded i-OCLs and CD4$^+$ T cells were seeded in a ratio 1:20 in IMDM medium supplemented with 5% FCS (Hyclone, GE Healthcare), 1% penicillin-streptomycin (Gibco) and 50 μM β-mercaptoethanol (Gibco). T cell proliferation was assessed by FACS after 3–5 days.

Implication of the PD-1/PD-L1 axis was evaluated by adding 5 μg/ml isotype control or anti-PD-1 (clone Rpm 14–1; kindly provided by V. Vouret (IRCAN, Nice, France)) antibodies at the beginning to the coculture. To validate the immune suppressive effect of Cx3cr1$^+$ i-OCLs, sorted Cx3cr1$^{neg}$ i-OCLs were stimulated with 600 nM OVA for 3 hr at 37°C, washed 3 times with IMDM medium and cultured together with CFSE-labelled CD4$^+$ T cells from transgenic OT-II mice on 96-well plates as described above. Cx3cr1$^+$ i-OCLs were sorted and added to the coculture without ovalbumin challenge in different ratios between Cx3cr1$^{neg}$ vs Cx3cr1$^+$ i-OCLs (1:1; 1:2; 2:1). T cell proliferation was assessed using FACS after 3–5 days of coculture.

## RNA sequencing on sorted osteoclasts

Total RNA (100 ng) from 4 biological replicates (from 4 different mice) in each group was extracted from in vitro differentiated i-OCLs (after sorting according to their multinucleation and GFP expression as described above and shown in *Figure 2—figure supplement 1*) with the RNeasy kit (Qiagen) and processed for directional library preparation using the Truseq stranded total RNA library kit (Illumina). Libraries were pooled and sequenced paired-ended for 2 × 75 cycles on a Nextseq500 sequencer (Illumina) to generate 30–40 million fragments per sample. After quality controls, data analysis was performed with 2 different approaches. For the first one, reads were 'quasi' mapped on the reference mouse transcriptome (Gencode vM15) and quantified using the SALMON software with the mapping mode and standard settings (*Patro et al., 2017*). Estimates of transcripts counts and their confidence intervals were computed using 1000 bootstraps to assess technical variance. Gene expression levels were computed by aggregating the transcript counts for each gene. Gene expression in biological replicates (n = 4) was then compared using a linear model as implemented in Sleuth (*Pimentel et al., 2017*) and a false discovery rate of 0.01. Lists of differentially expressed genes were annotated using Innate-DB and EnrichR web portals. For the second approach, raw RNAseq fastq reads were trimmed with Trimmomatic and aligned to the reference mouse transcriptome (Gencode mm10) using STAR (v. 2.6.1 c) (*Dobin et al., 2013*) on the National Institutes of Health high-performance computing Biowulf cluster. Gene-assignment and estimates counts of RNA reads were performed with HTseq (*Anders et al., 2015*). Further analyses were performed with R software and gene expression in biological replicates (n = 4) was compared between the different conditions to identify differentially expressed genes using DESeq2 (*Love et al., 2014*) with the Wald test (FDR < 0.01). Batch removal was performed using limma (*Ritchie et al., 2015*). Gene ontology (GO) pathway analyzes were performed using Goseq (*Young et al., 2010*). Both approaches gave equivalent results (not shown).

## Gene expression analyses

Total RNA of sorted in vitro differentiated Cx3cr1$^+$ and Cx3cr1$^{neg}$ i-OCLs was extracted using TRIzol reagent with subsequent isopropanol precipitation following manufacturer's instructions. RT-qPCR was performed after reverse transcription (Superscript II, Life Technologies) as described previously (*Mansour et al., 2012*) using SYBR Green and the primers indicated in the Key ressources table. Samples of 3 biological replicates were run in triplicates and results were normalized to the reference gene *36B4* using the $2^{-\Delta Ct}$ method as described (*Mansour et al., 2011*).

## Statistical analyses

All data were analyzed using Graph Pad Prism 7.0 software using an appropriate two tailed student's t-test with Bonferroni adjustment when comparing two groups. When more than two groups were compared two-way analysis of variance (ANOVA) was used. Statistical significance was considered at p<0.05. Experimental values are presented as mean ± standard deviation (SD) of at least three biological replicas. Error bars for gene expression analysis of humans and mice using RT-qPCR show the mean value with 95% confidence interval. All experiments were repeated with a minimum of three biological replicates and at least two technical replicates.

## Acknowledgements

We acknowledge M Topi (LP2M, Nice, France) for her technical assistance, the Genomic Facility of the UFR Simone Veil, (Université Versailles-Saint-Quentin, France) for the RNA sequencing, the IRCAN animal core facility (Nice, France) and the preclinical platform of ECELLFRANCE for µCT analysis (IRMB, Montpellier, France). This work utilized the computational resources of the NIH-HPC-Biowulf cluster (http://hpc.nih.gov).

The work was supported by the Agence Nationale de la Recherche (ANR-16-CE14-0030), the French government, managed by the ANR as part of the Investissement d'Avenir UCA^JEDI project (ANR-15-IDEX-01), the Fondation Arthritis, Société Française de Biologie des Tissus Minéralisés, European Calcified Tissue Society and American Society of Bone and Mineral Research. M-B M is supported by the Fondation pour la Recherche Médicale (FRM, ECO20160736019) and TC is supported by the Intramural Research Program of the National Cancer Institute, Center for Cancer Research, National Institutes of Health.

## Additional information

### Funding

| Funder | Grant reference number | Author |
| --- | --- | --- |
| Agence Nationale de la Recherche | ANR-16-CE14-0030 | Florence Apparailly Henri-Jean Garchon Claudine Blin-Wakkach |
| Agence Nationale de la Recherche | ANR-15-IDEX-01 | Maria-Bernadette Madel Claudine Blin-Wakkach |
| Fondation pour la Recherche Médicale | ECO20160736019 | Maria-Bernadette Madel |
| Fondation Arthritis | | Maria-Bernadette Madel Claudine Blin-Wakkach |
| Société Française de Biologie des TissusMinéralisés | | Maria-Bernadette Madel |
| European Calcified Tissue Society | | Maria-Bernadette Madel |
| American Society of Bone and Mineral Research | | Maria-Bernadette Madel |
| National Cancer Institute | Intramural Research Program | Thomas Ciucci |

The funders had no role in study design, data collection and interpretation, or the decision to submit the work for publication.

### Author contributions

Maria-Bernadette Madel, Formal analysis, Validation, Investigation, Visualization, Methodology, Writing - original draft, Writing - review and editing; Lidia Ibáñez, Formal analysis, Validation, Investigation, Methodology, Writing - review and editing; Thomas Ciucci, Formal analysis, Visualization, Writing - review and editing; Julia Halper, Formal analysis, Investigation, Methodology; Matthieu Rouleau, Visualization, Writing - review and editing, contributed to the discussion of results and provided helpful advices throughout the study; Antoine Boutin, Formal analysis, Validation, Investigation; Christophe Hue, Data curation, Validation, Investigation, Methodology; Isabelle Duroux-Richard, Formal analysis, Writing - review and editing; Florence Apparailly, Supervision, Writing - review and editing, Contributed to the discussion of results and provided helpful advices throughout the study; Henri-Jean Garchon, Formal analysis, Validation, Visualization, Methodology, Writing - review and editing; Abdelilah Wakkach, Conceptualization, Supervision, Visualization, Methodology, Writing - review and editing; Claudine Blin-Wakkach, Conceptualization, Formal analysis, Supervision, Funding acquisition, Validation, Visualization, Methodology, Writing - original draft, Project administration, Writing - review and editing

## Author ORCIDs

Matthieu Rouleau [iD] http://orcid.org/0000-0002-0075-9880
Claudine Blin-Wakkach [iD] https://orcid.org/0000-0002-2621-3907

## Ethics

Animal experimentation: The protocol for animal experiments was approved by the Institutional Ethics Committee on Laboratory Animals (CIEPAL-Azur, France) (#2016121216457153 (V3). Experiments were conducted in compliance with ethical regulations for animal testing and research.

## Decision letter and Author response

Decision letter https://doi.org/10.7554/eLife.54493.sa1
Author response https://doi.org/10.7554/eLife.54493.sa2

## Additional files

### Supplementary files

• Source data 1. Raw counts-HTSeq. Raw counts (HTseq) per gene and per sample from the RNAseq analysis of Cx3cr1$^+$ and Cx3cr1$^{neg}$ i-OCLs from WT $Cx3cr1^{GFP/+}$ mice and Cx3cr1$^+$ i-OCLs from deficient $Cx3cr1^{GFP/GFP}$ mice. In each group, the 4 i-OCL samples have been obtained from 4 different mice.

• Source data 2. Counts_Deseq NormaBatch Corrected. Counts per gene and per sample after normalization and batch correction from the RNAseq analysis of Cx3cr1$^+$ and Cx3cr1$^{neg}$ i-OCLs from WT $Cx3cr1^{GFP/+}$ mice and Cx3cr1$^+$ i-OCLs from deficient $Cx3cr1^{GFP/GFP}$ mice. In each group, the 4 i-OCL samples have been obtained from 4 different mice.

• Source data 3. Cx3cr1$^+$ i-OCLs from WT $Cx3cr1^{GFP/+}$ mice versus Cx3cr1$^+$ i-OCLs from deficient $Cx3cr1^{GFP/GFP}$ mice _Deseq. Statistics (Deseq) for the comparison of expressed genes in Cx3cr1$^+$ i-OCLs from WT $Cx3cr1^{GFP/+}$ mice versus Cx3cr1$^+$ i-OCLs from deficient $Cx3cr1^{GFP/GFP}$ mice. In each group, the 4 i-OCL samples have been obtained from 4 different mice.

• Source data 4. Cx3cr1$^+$ i-OCLs from WT $Cx3cr1^{GFP/+}$ mice versus Cx3cr1$^{neg}$ i-OCLs from WT $Cx3cr1^{GFP/+}$ mice mice _Deseq. Statistics (Deseq) for the comparison of expressed genes in Cx3cr1$^+$ versus Cx3cr1$^{neg}$ i-OCLs from WT $Cx3cr1^{GFP/+}$ mice. In each group, the 4 i-OCL samples have been obtained from 4 different mice.

• Transparent reporting form

## Data availability

All the RNA sequencing data are included in the submitted manuscript as data source files. RNA-Sequencing data have been deposited in ENA (European Nucleotide Archive) under accession number PRJEB36092.

The following dataset was generated:

| Author(s) | Year | Dataset title | Dataset URL | Database and Identifier |
|---|---|---|---|---|
| Madel MB, Garchon HJ, Wakkach A, Blin-Wakkach C | 2020 | Heterogeneity of inflammatory osteoclasts based on CX3CR1 expression | https://www.ebi.ac.uk/ena/data/view/PRJEB36092 | European Nucleotide Archive, PRJEB36092 |

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
