## [Decision Letter]

**Acceptance summary:**

The authors studied the heterogeneity of inflammatory osteoclasts (i-OCL) using Cx3cr1 as a marker of osteoclast heterogeneity in a mouse model of OVX-induced bone resorption. Cx3cr1-neg OCLs showed higher capacity for bone matrix dissolution, phagocytosed less OVA antigen, and had higher capacity to induce T cell proliferation in vitro, compared to Cx3cr1+ OCLs.

**Decision letter after peer review:**

Thank you for submitting your article "Dissecting the phenotypic and functional heterogeneity of inflammatory osteoclasts by the expression of CX3CR1" for consideration by *eLife*. Your article has been reviewed by three peer reviewers, and the evaluation has been overseen by a Reviewing Editor and Kathryn Cheah as the Senior Editor. The reviewers have opted to remain anonymous.

The reviewers have discussed the reviews with one another and the Reviewing Editor has drafted this decision to help you prepare a revised submission.

Essential revisions:

1) The conclusions rely on (1) gene expression profiling of sorted CX3CR1+ and CX3CR1- cells, and (2) in vitro culture followed by functional analysis of said sorted cells. However, for these in vitro experiments, the culture conditions employed prior to analysis are not clear. The Materials and methods section suggests that there was extensive culture and stimulation prior to analysis following sorting, but these conditions are not explicitly stated in the main text. If cells were extensively cultured with stimulating cytokines prior to functional analysis, then this would be expected to drastically change their functional profile compared to cells sorted directly from the in vivo environment. This would affect the results in Figures 2, 4, 5, and 6.

2) While Cx3cr1+ and Cx3cr1- cells have different effects on T cell proliferation, that doesn't necessarily mean that one regulates the activity of the other in vivo. To make this conclusion it would be necessary to deplete selectively one population in vivo and observe the effects on the other population.

3) Given that macrophages can fuse into multinucleated cells and that Cx3cr1 is a marker of macrophage heterogeneity, it has not been demonstrated that the cells were indeed OCLs and not macrophages. Evidence of the cell identity, OCL or macrophage should be provided.

4) The absence of Cx3cr1 in Cx3cr1 gfp/gfp mice resulted in reduced recruitment of dendritic cells and Ly6C-hi monocytes to the sites of bone resorption in OVX animals. This result would be expected because Cx3cr1 is well known to be involved in leukocyte trafficking. The authors concluded that this result shows that Ly6C-hi monocytes are progenitors of inflammatory OCLs. This conclusion is not supported by the results.

5) The authors conclude that Cx3cr1+ cells express higher levels of immunosuppressive genes compared to Cx3cr1- cells. Because the list of genes clearly includes many genes that are immunostimulatory (IL1b, CCL5, Nfkb1, Ccl2, Cd80, CD86), the conclusions about immunosuppressive gene expression profile in Cx3cr1- cells are not clearly supported.

6) In Figure 1 the authors claim that CXC3R1 positive osteoclasts are needed for the full bone resorbing activity of osteoclasts after ovariectomy. The claim is based on only minor differences and whether CXC3R1 positive osteoclasts were preset or not, the osteoporosis phenotype develops in an equally severe manner. Moreover, these data contradict a statement made later in the paper that CXC3R1 positive osteoclasts are not needed for bone resorption.

7) Data shown in Figure 2 (no difference in the bone resorption capacities of CX3CR1+ cells isolated from GFP/+ mice and CX3CR1- cells isolated from GFP/GFP mice) and those shown in Figure 4 (large differences in bone resorptive capacities of CX3CR1+ and CX3CR1-negative cells derived from dendricytes) are contradictory. This inconsistency calls into question the validity of using cells derived from the dendricytes in culture if they behave differently from the mature cells isolated from the genetically modified mice.

---

## [Author Response]

Essential revisions:1) The conclusions rely on (1) gene expression profiling of sorted CX3CR1+ and CX3CR1- cells, and (2) in vitro culture followed by functional analysis of said sorted cells. However, for these in vitro experiments, the culture conditions employed prior to analysis are not clear. The Materials and methods section suggests that there was extensive culture and stimulation prior to analysis following sorting, but these conditions are not explicitly stated in the main text. If cells were extensively cultured with stimulating cytokines prior to functional analysis, then this would be expected to drastically change their functional profile compared to cells sorted directly from the in vivo environment. This would affect the results in Figures 2, 4, 5, and 6.

The reviewers raised an important point. We generated in vitro inflammatory OCLs in the presence of RANK-L and M-CSF, the 2 main osteoclastogenic factors, according to the gold standard protocol for in vitro osteoclastogenesis and starting from their known progenitors, BM-derived dendritic cells (Ibanez et al., 2016). Of course, we agree that investigating osteoclast function on *ex-vivo* isolated osteoclasts would have been the best approach. However, up to now, there are no publications investigating osteoclast function on murine ex vivo-isolated osteoclasts. These cells are rare and firmly attached to the bone and to date there's no procedure in the literature to isolate them in sufficient number to perform subsequent analysis. Consequently, functional assays on osteoclasts always use in vitro-generated cells and much of osteoclast biology described so far in the literature has been investigated in such conditions (Madel et al., 2018). Moreover, at the moment there is no marker to distinguish between inflammatory and tolerogenic Cx3cr1^neg^ osteoclasts, which make it impossible to sort them ex vivo. Up to know, the only way to obtain these cells is to differentiate them from their respective known progenitors. Of course, we agree with the reviewers that this may affect their function, but to the same extend as in all studies investigating osteoclast activity and published so far and we don't have any other option.

To make the procedure of osteoclast preparation clearer as requested, we revised the main text and included more details in the Results and Materials and methods sections as well as in the figure legends. We also added in Figure 2 and Figure 3 a schematic representation of osteoclast preparation.

2) While Cx3cr1+ and Cx3cr1- cells have different effects on T cell proliferation, that doesn't necessarily mean that one regulates the activity of the other in vivo. To make this conclusion it would be necessary to deplete selectively one population in vivo and observe the effects on the other population.

The reviewers are perfectly right and demonstrating the interaction between the 2 i-OCLs subsets in vivo would require specific depletion in vivo of one of these subsets. We are currently investigating in more details the heterogeneity of the different OCL subsets in order to identify suitable factors that would allow to specifically target one or the other OCL subset. This study, and in particular the validation of markers, requires time and the data will be published in a future manuscript.

However, to address the reviewer's concern and to avoid any over-interpretation of our results, in the revised manuscript we softened our conclusion and clearly stated that the mechanisms we described are only shown in vitro and remain to be demonstrated in vivo.

3) Given that macrophages can fuse into multinucleated cells and that Cx3cr1 is a marker of macrophage heterogeneity, it has not been demonstrated that the cells were indeed OCLs and not macrophages. Evidence of the cell identity, OCL or macrophage should be provided.

To prove the identity of the cells we are analyzing, we performed additional experiments as requested. In particular, because in our protocol osteoclasts are generated in the presence of RANK-L but also M-CSF, a potent inducer of macrophages, we compared these osteoclasts to macrophages generated from bone marrow cells in the presence of M-CSF. In these conditions, we didn't observed cell fusion in macrophages. Moreover, our results showed that while BM-derived macrophages express the 2 BM macrophage markers CD68 and CD169 but are negative for TRAcP expression (Figure 4—figure supplement 1), both Cx3cr1^+^ and Cx3cr1^neg^ subsets of OCLs do not express these macrophage markers but share osteoclast hallmarks, such as TRAcP expression and resorption capacity (Figure 4C-F). These data were included in the revised manuscript and confirm that the cells we are analyzing are *bona fide* OCLs, whether they express or not Cx3cr1.

4) The absence of Cx3cr1 in Cx3cr1 gfp/gfp mice resulted in reduced recruitment of dendritic cells and Ly6C-hi monocytes to the sites of bone resorption in OVX animals. This result would be expected because Cx3cr1 is well known to be involved in leukocyte trafficking. The authors concluded that this result shows that Ly6C-hi monocytes are progenitors of inflammatory OCLs. This conclusion is not supported by the results.

Ly6C^hi^ monocytes are involved in osteoclastogenesis in particular in conditions related to inflammatory bone destruction (Seeling et al., 2013; Ammari et al., 2018; Madel et al., 2019), which suggest that they could be progenitors of Cx3cr1^+^ osteoclasts that are also associated to the same conditions (Ibanez et al., 2016). To address the capacity of Ly6C^hi^ monocytes to differentiate into Cx3cr1^+^ osteoclasts, we sorted Ly6C^hi^ monocytes from the blood of WT *Cx3cr1*^GFP/+^ mice and cultured them with M-CSF and RANKL for 6 days as shown in Figure 1—figure supplement 1A. We analyzed the resulting osteoclasts by FACS and found that about 35% of these osteoclasts are GFP/Cx3cr1^+^ (Figure 1—figure supplement 1B) demonstrating that Ly6C^hi^ monocytes are indeed progenitors of Cx3cr1^+^ osteoclasts, which support our conclusion. To make this point more visible and clearer in the revised version of the manuscript, we modified the text in the Results section and added a schematic representation of the experiment (Figure 1—figure supplement 1A).

5) The authors conclude that Cx3cr1+ cells express higher levels of immunosuppressive genes compared to Cx3cr1- cells. Because the list of genes clearly includes many genes that are immunostimulatory (IL1b, CCL5, Nfkb1, Ccl2, Cd80, CD86), the conclusions about immunosuppressive gene expression profile in Cx3cr1- cells are not clearly supported.

There might be a typo error (Cx3cr1- instead of Cx3cr1+) in the last sentence of the reviewer's comment because we only discussed the immunosuppressive capacity of Cx3cr1^+^ cells, not of Cx3cr1^neg^ cells.

The reviewers are perfectly right and Cx3cr1^+^ osteoclasts do not only express high levels of immunosuppressive genes but also of genes involved in inflammation (Figure 6A), which is in line with their capacity to stimulate to some extend the proliferation of inflammatory CD4^+^ T cell (Figure 5G, Figure 5—figure supplement 1). Thus, we agree that just the RNA expression profile of these OCLs is not fully supporting their immunosuppressive role. In the revised manuscript, the comment on the RNAseq analysis for this figure in the Results section has been rephrased accordingly. However, our conclusions do not only rely on the expression profile of osteoclasts, but also on in vitro functional analysis that clearly show:

1) that the T cell activation capacity of Cx3cr1^+^ OCLs is increased when blocking the immunosuppressive checkpoint protein PD1 in vitro (Figure 6C), 2) that in an immunosuppressive assay, Cx3cr1^+^ OCLs reduce the capacity of Cx3cr1^neg^ OCLs to stimulate T cell proliferation (Figure 6D-E) which is a clear functional demonstration of their immunosuppressive effect, at least in vitro as mentioned in the revised version.

6) In Figure 1 the authors claim that CXC3R1 positive osteoclasts are needed for the full bone resorbing activity of osteoclasts after ovariectomy. The claim is based on only minor differences and whether CXC3R1 positive osteoclasts were preset or not, the osteoporosis phenotype develops in an equally severe manner. Moreover, these data contradict a statement made later in the paper that CXC3R1 positive osteoclasts are not needed for bone resorption.

We agree that the differences observed when comparing OVX WT mice and OVX Cx3cr1-deficient mice (Figure 1B-C) for the microscanner analysis are minor but they are significant and are further supported by a reduced osteoclast number (Figure 1D). But to follow the reviewer concern, we rephrased the comment of this figure in the Results section to indicate this low degree of variation.

However, there might be a misunderstanding as we didn't claim that Cx3cr1^+^ OCLs are required for bone loss in OVX mice. Figure 1 is not addressing the role of Cx3cr1^+^ osteoclasts but the implication of the Cx3cr1 protein in bone destruction associated to ovariectomy, which are 2 different questions. In Figure 1, we showed that Cx3cr1, as a protein, is important for the recruitment of OCL progenitors in the bone marrow of OVX mice (Figure 1E-F). Moreover, we showed that i-OCLs from Cx3cr1-deficient and from WT mice have the same resorption activity (Figure 2C), which demonstrates that the protein Cx3cr1 is not important for this activity.

Moreover, we didn't claim that Cx3cr1^+^ OCLs are not needed for bone resorption, we only showed that they resorb less than the Cx3cr1^neg^ OCLs (Figure 4E-F).

But to avoid any misunderstanding, we modified the text to make it clearer and to describe in more details the mouse model we used. We also added in Figure 2 and 3 a schematic representation of the preparation of the cells to avoid any confusion.

7) Data shown in Figure 2 (no difference in the bone resorption capacities of CX3CR1+ cells isolated from GFP/+ mice and CX3CR1- cells isolated from GFP/GFP mice) and those shown in Figure 4 (large differences in bone resorptive capacities of CX3CR1+ and CX3CR1-negative cells derived from dendricytes) are contradictory. This inconsistency calls into question the validity of using cells derived from the dendricytes in culture if they behave differently from the mature cells isolated from the genetically modified mice.

As explained in the first point above, due to methodological issues, in particular the absence of a protocol to isolate sufficient osteoclasts or to sort ex vivo specific subsets of OCLs, there is no other possibility to investigate the function of these OCLs than the use of in vitro differentiated cells. In particular in our study, all inflammatory osteoclasts used in functional assays were derived from dendritic cells in exactly the same culture conditions, as already described (Ibanez et al., 2016). Thus, even if we cannot exclude that the culture conditions could affect osteoclast function compared to in vivo, it would be to the same extend in all our analysis. It would also be to the same extend as in all studies investigating osteoclast activity published so far.

Regarding the reviewer's concern on potential inconsistency in our results, there might be a misunderstanding. In Figure 2, we addressed the role of Cx3cr1, as a protein, in bone resorption and T cell activation by osteoclasts. This was achieved comparing Cx3cr1^+^ osteoclasts coming either from WT (Cx3cr1^GFP/+^) mice or from mice baring a non-functional Cx3cr1 gene (Cx3cr1^GFP/GFP^). There are no data on Cx3cr1^neg^ cells in Figure 2. And indeed, we showed that Cx3cr1^+^ cells have equivalent resorption capacity whether Cx3cr1 is functional or not (Figure 2), which demonstrates that the protein Cx3cr1 is not important for this activity.

In contrast, in Figure 4, we addressed, in WT mice, the specific function of the 2 distinct subsets of osteoclasts that are identified using Cx3cr1 as a membrane marker. We showed that Cx3cr1^+^ and Cx3cr1^neg^ i-OCLs from WT mice differ in their resorption and T cell activation capacity (Figures 3-6).

Thus, Figures 1-2 and Figures 3-6 explore 2 completely distinct aspects that could not be compared.

Importantly, Cx3cr1^+^ OCLs having a non-functional *Cx3cr1* gene could not be considered equivalent to Cx3cr1^neg^ OCLs from WT mice. Indeed, as shown in Author response image 1, principal component analysis of our transcriptomic data clearly demonstrated that while Cx3cr1^+^ i-OCLs from WT and mutant mice are closely related, both differ considerably from Cx3cr1^neg^ cells from WT mice as they cluster in 2 completely distinct groups (panel A). This was further confirmed by matrix similarity analysis of our transcriptomic data that showed again that the 2 populations are distinct (panel B). Therefore none of the conclusions made for one of these population can be applied to the other.

However, we recognize that switching from the analysis of the implication of the Cx3cr1 protein in bone loss to the analysis of the respective role of Cx3cr1^+^ and Cx3cr1^neg^ i-OCL could be confusing. Thus, in the revised manuscript we modified the text to make it clearer and we added schematic representations of the preparation of OCLs in the figures.

**Author response image 1. sa2fig1:** Cx3cr1^neg^ i-OCLs from Cx3cr1^GFP/+^ mice are distinct from Cx3cr1^+^ i-OCLs from deficient Cx3cr1^GFP/GFP^ mice. A) Principal component analysis of Cx3cr1^+^ from deficient Cx3cr1^GFP/GFP^ mice (green) and Wt Cx3cr1^GFP/+^ mice (red) and Cx3cr1^neg^ i-OCLs from Cx3cr1^GFP/+^ mice (blue). Data shows the two first components after batch correction. Each dot represents the expression profile of one sample. B) Similarity matrix analysis on the same samples.